# Calibration of cognitive tests to address the reliability paradox for decision-conflict tasks

Talira Kucina ®[1] ✉, Lindsay Wells ®[2], Ian Lewis[2], Kristy de Salas[2], Amelia Kohl[1], Matthew A. Palmer[1], James D. Sauer[1], Dora Matzke[3], Eugene Aidman[4,5] & Andrew Heathcote[3,6]

Standard, well-established cognitive tasks that produce reliable effects in group comparisons also lead to unreliable measurement when assessing individual differences. This reliability paradox has been demonstrated in decision-conflict tasks such as the Simon, Flanker, and Stroop tasks, which measure various aspects of cognitive control. We aim to address this paradox by implementing carefully calibrated versions of the standard tests with an additional manipulation to encourage processing of conflicting information, as well as combinations of standard tasks. Over five experiments, we show that a Flanker task and a combined Simon and Stroop task with the additional manipulation produced reliable estimates of individual differences in under 100 trials per task, which improves on the reliability seen in benchmark Flanker, Simon, and Stroop data. We make these tasks freely available and discuss both theoretical and applied implications regarding how the cognitive testing of individual differences is carried out.

Differential psychology aims to measure the way in which individuals vary in their behavior and to understand the processes that cause those differences. These causes are conceptualized as stable individual characteristics, or traits, but the behavioral effects of traits on any given measurement occasion are modulated by fluctuations in transient states, such as arousal, attention, mood, fatigue, and the intrinsic variability in the nervous system. State variation acts as a type of measurement noise with respect to the correlation-based techniques that are used in differential psychology to estimate the proportion of behavior that can be explained by trait variation. It has long been acknowledged that correlations based on averages over measurement occasions under-estimate this proportion[1], leading to an emphasis on the development of reliable scales with levels of measurement noise that are small relative to the underlying level of trait variation. In this paper, we focus on the reliable measurement of individual differences in the ability to control the decision conflict caused by interference from misleading or irrelevant information. Decision conflict is apparent in many real-world high-stakes contexts, such as shoot/don't shoot

scenarios, where strong inhibitory skills ensure superior performance[2]. The ability to inhibit conflicting information is recognized as a key component of individual differences in areas ranging from executive-control[3,4] to aging[5].

Individual differences in conflict control are often measured using choice tasks, such as the Flanker, Simon, or Stroop. In such tasks, the conflict effect is traditionally measured by the difference in mean response time (RT) between an incongruent condition, where information from an irrelevant stimulus attribute conflicts with information from another stimulus attribute that is relevant for the choice, and a control condition, where the two sources of information are congruent, or the irrelevant attribute is absent (e.g., Fig. 1). The irrelevant attributes are chosen so they activate cognitive representations that interfere with response selection. This can occur because of pre-potent factors, including the tendency to respond based on reading a color word in the Stroop task instead of based on the color the word is printed in, or responding towards a stimulus position in the Simon task rather than the rule mapping stimulus color to a response button on the side

[1]School of Psychological Sciences, University of Tasmania, Hobart, TAS, Australia. [2]Games and Creative Technologies Research Group, University of Tasmania, Hobart, TAS, Australia. [3]Department of Psychology, University of Amsterdam, Amsterdam, The Netherlands. [4]Defence Science Technology Group, Canberra, NSW, Australia. [5]School of Biomedical Sciences & Pharmacy, University of Newcastle, Newcastle, NSW, Australia. [6]School of Psychology, University of Newcastle, Newcastle, NSW, Australia. ✉e-mail: talira.kucina@utas.edu.au

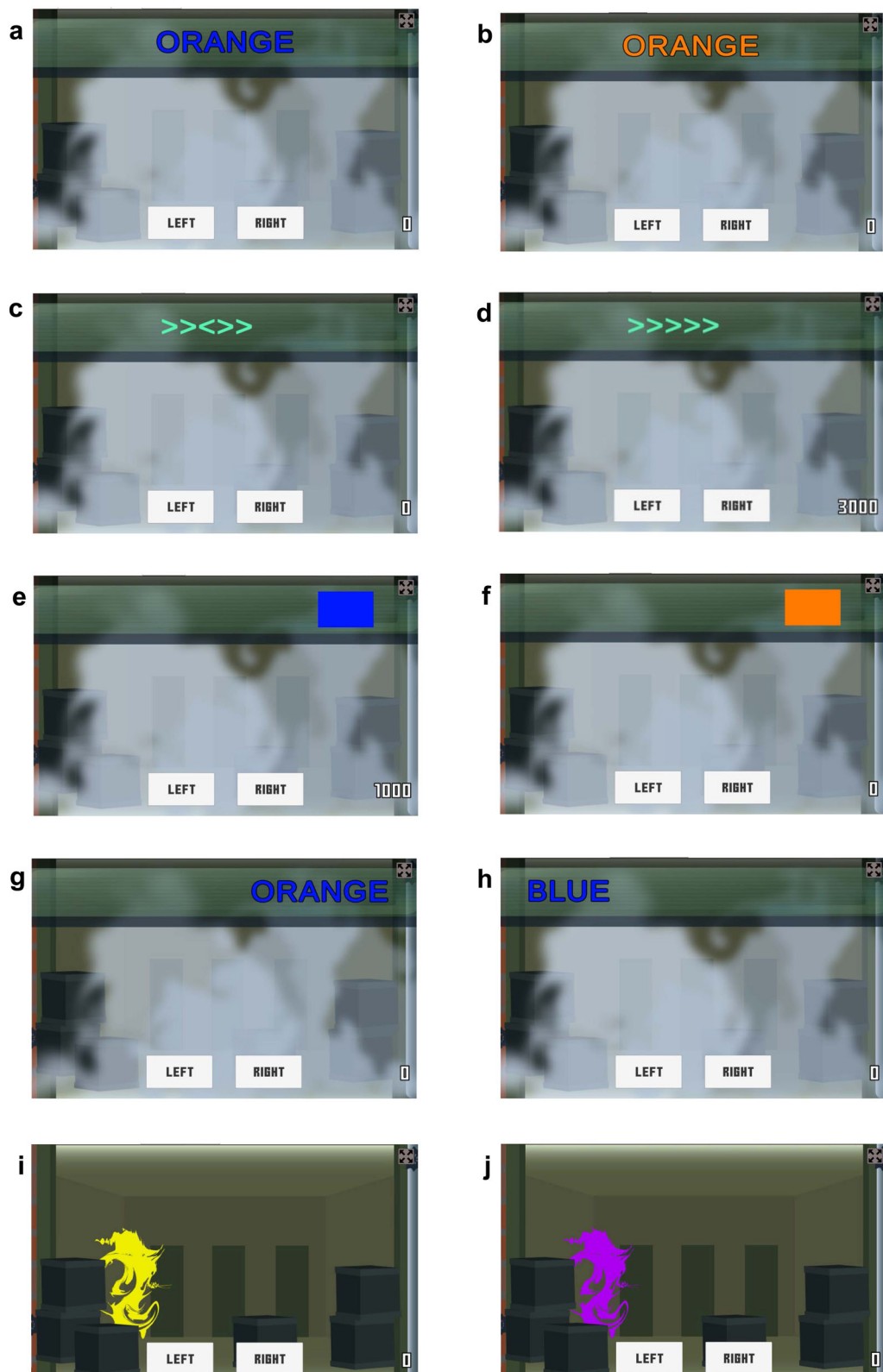

opposite the stimulus. In the Flanker task, fine-grained spatially-based attention selection is required between the relevant (target) and irre-levant (distractor) characteristics due to high similarity (e.g., left- vs. right-facing arrows), differing only slightly in terms of their spatial location (central for the target with adjacent distractors nearby).

RT difference scores have high face validity as measures of the specific type of interference control each task requires (i.e., of different pre-potent tendencies in the Stroop and Simon tasks, and of spatial attention in the Flanker task). Taking the difference between incon-gruent and control RT accounts for the effect of variation in factors such as overall speed that could otherwise confound the measurement of trait differences in conflict control. In the experimental psychology tradition in which these tasks were developed, they produce highly robust RT differences in condition averages over participants.

**Fig. 1 | Illustrations of the tasks used in the present experiments.** The stimuli on which decisions are based are presented in a head-ups display at the top of the screen. Stroop stimuli are colored words, the relevant information is stimulus color (e.g., press the left key for blue and right for orange) and the irrelevant attribute is word meaning: (**a**) incongruent Stroop display, and (**b**) congruent Stroop display. Flanker stimuli are sets of left or right-pointing arrows, where the response is based on the direction of the central arrow and the irrelevant information is from closely flanking arrows: (**c**) incongruent Flanker display, and (**d**) congruent Flanker display. The Simon task puts an arbitrary stimulus-response mapping rule (press the left key for a blue rectangle and right for a orange rectangle) in conflict with stimulus location (e.g., incongruent = blue rectangle presented on the right): (**e**) incongruent Simon display and, (**f**) congruent Simon display. In the combined Stroop and Simon (Stroopon) task, responses are based on word color, with words presented on the left or right of the screen: (**g**) double incongruent Stroopon display and, (**h**) double congruent Stroopon display. After a response based on the initial stimulus the enemy appears and feedback is given except on double-shot trials where the enemy appears behind a colored shield. The final two panels show second shot Stroopon displays following a response to the display in (**g**) or (**h**): (**i**) the yellow-colored shield requires a response based on location, and (**j**) a purple-colored shield requires a response based on the word.

The validity of RT differences have been repeatedly demonstrated in experimental studies showing that their magnitude is modulated by experimental manipulations of the specific types of control they require (for reviews of each task, see Eriksen[6], Hommel[7], and MacLeod[8]). Converging support for their validity has been provided by the neurosciences, where these tasks have formed a basis for identifying the brain areas mediating different types of cognitive control[9-11].

Despite long-held aspirations[12] and progress[13], in bringing together experimental and differential disciplines, the correlation-based analyses used in the latter can be problematic when applied to tasks from experimental psychology. Robust experimental effects are not always associated with robust individual difference correlations[14], and the compounding of measurement noise caused by taking RT differences[15], decreases the reliability of correlation-based analyses[3,16]. This issue was recently coined the reliability paradox[17] and illustrated in the conflict tasks presented here. In the next sections we discuss why this unreliability occurs and how it can be quantified, and then propose various types of conflict tasks with improved reliability.

Reliability ($r$) is the ratio of trait, or true score, variation, $\sigma_T^2$ (i.e., the variance of individual differences in the trait), to the total variation in behavior, which is the sum of trait variance and state variance, $\sigma_S^2$:

$$r = \sigma_T^2 / (\sigma_T^2 + \sigma_S^2) \tag{1}$$

Traditional approaches based on effects averaged over test trials do not afford direct assessment of the components (i.e., state and trait variation) that make up reliability. Instead, it is assessed through techniques such as split-half correlation (between odd vs. even trials) or test-retest correlation (between performance on two separate testing sessions). Modern Bayesian analyses based on trial-level data, recently introduced to neuroscience[18] and cognitive psychology[19], directly estimate these variances and the uncertainty with which they are estimated. Rouder and colleagues[20,21] applied this approach to conflict data, using hierarchical models to estimate $\sigma_T$ and the measurement noise associated with trial-to-trial performance variations, $\sigma_N$. State variance, $\sigma_S^2$, is equal to the squared standard error of the conflict effect, thus, it increases with noise variance and decreases with the number of test trials. Assuming both the control and incongruent conditions have $L$ trials each:

$$\sigma_S^2 = 2\sigma_N^2 / L \tag{2}$$

Findings with respect to the number of trials required to achieve reliable measurement of decision-conflict control are bleak[21]: across a large sample of conflict studies, the median ratio of trait (i.e., conflict effect) to noise standard deviations equaled $\eta = \sigma_T/\sigma_N = 0.13$ (for fMRI measurements the situation is worse[22], $\eta \sim 0.05$). Larger values of $\eta$ (trait precision) increase the precision with which trait variability is measured. Therefore, measuring conflict with reliability $r$ requires:

$$L = 2r^2 / (\eta^2 (1 - r^2)) \tag{3}$$

Hence, where $\eta = 0.13$, $2L = 420$ test trials in total are needed to reach a conventionally accepted level of adequate reliability, $r = 0.8$ (note L is

doubled to account for the two conditions contributing to the RT difference, incongruent and control).

Given that many applications of decision-conflict tasks in differential psychology are part of a large battery, $L$ is usually small. For example, Friedman and Miyake[3] report a split-half reliability of 0.59 with $L = 40$ Flanker RT difference scores, consistent with low trait precision ($\eta = 0.16$), making the utility of such scores questionable. Taking a difference might also remove some other sources of individual variability in cognitive control. For example, it has been argued that even non-conflict choices require a degree of cognitive control that reliably varies between individuals[23,24], and so measurement of this type of control would be compromised by taking differences. Similarly, relationships between measures of working memory capacity and RT in congruent conditions of the Stroop[25] and Flanker[26] tasks may be indicative of this condition requiring executive-control processes, although the relationship between working memory capacity and executive-control is debated[24].

Such considerations have prompted calls to abandon measures based on conflict-task RT difference scores in differential research[27], and instead center on either the development of alternative measurement approaches[28], or shift focus to differences in processing speed and strategy[29]. While such approaches are welcome, entirely abandoning RT difference measures would forgo their well-established validity as measures of control of the specific types of interference present in tasks such as the Flanker, Stroop, and Simon. We acknowledge that the specificity afforded by RT difference scores may come at the cost of compromising the measurement of other forms of executive control. However, we argue that the clear importance of these particular types of interference makes it desirable to develop reliable tasks. Practically, reliability is a pre-requisite for using such tasks to select individuals who are likely to excel in situations subject to specific types of interference and conflict. Theoretically, reliability is required to investigate the controversial topic of whether there are domain-general types of executive control[30]. Further, it is important to note that the magnitude of correlations among conflict-task RT difference scores, which have been central to such investigations, do not bear on the question of their validity. Indeed, it has been argued that domain-general concepts including inhibition and updating are so ill-defined that they should be replaced by developing mechanistic explanations in terms of targeting control at one or more of the signal detection, action selection, and action execution stages of processing[31]. For all these purposes, reliable conflict-task RT difference scores are desirable, thus we explore avenues for improvement.

Over a series of online experiments, we refined the design of Flanker, Simon, and Stroop tasks to increase a key ingredient of reliability, the magnitude of conflict effects. The first set of refinements targeted the signal-detection stage. Displays were constructed so irrelevant information was salient. That is, Simon task stimuli were presented at the far sides of the screen and other tasks used large, legible characters. To make it difficult for participants to preemptively focus their visual attention on the target in the Flanker task, characters were closely spaced and randomly jittered by 0–2 character-spaces, thus, a target could appear in any position occupied by a distractor when there was no jitter. See Fig. 1 for task examples.

Larger conflict effects also occur when responses based on irrelevant attributes are sometimes required[32,33], presumably because participants must complete the signal detection stage for the irrelevant attribute. We compared Flanker, Simon, and Stroop tasks—with only the standard single response—to double-shot versions (designated Flanker2, Simon2 etc.). A randomly selected 1/3 of trials required a double-shot, where after a correct standard response, a second response based on the irrelevant attribute was required. Table 1 indicates which experiments and tasks required single and double-shot responses. Taking the Stroop task as an example (Fig. 2), an initial response is based on stimulus color, then the second response is dependent upon reading the word. Accurate second responses meant that participants did not adopt signal-detection stage interference control strategies like blurring the eyes to stop reading in the Stroop task.

Next, each task used a video game format that made the tasks more demanding, yet also more engaging, than typical versions. The gamified tasks followed a storyline where participants responded to an enemy hidden behind boxes on either the left or right of the display, with their location indicated by a display at the top of the screen (e.g., right-pointing target arrow in Flanker indicated an enemy on the right). After the first response on standard trials, the enemy appeared, and feedback provided. On double-shot trials the enemy then appeared in a different color, indicating they had raised a shield and a second response based on the irrelevant attribute was required (see Figs. 1, 2). Disengagement is undesirable if it leads to strategies like careless or random responding that increase measurement noise, which could be problematic particularly if more trials are used to increase reliability, ultimately leading to fatigue. Greater complexity could also increase the effects of individual differences in attention capacity. For example, in prospective-memory paradigms requiring cognitive control[34], greater overall task complexity causes the emergence of limited attention capacity effects that are not evident in simpler versions of the task[35]. The double-shot manipulation might act similarly because it requires multiple response rules to be held in working memory.

Finally, we combined Stroop and Simon tasks (resulting in Stroopon) and Flanker and Simon tasks (resulting in Flankon), with the aim of obtaining a larger combined conflict effect. Flanker or Stroop stimuli were presented to the left or right of the display (see Fig. 1). Such task combinations might prove to be a simple way of increasing reliability. However, we return to the implications of combining tasks for the type of interference being measured in the Discussion. Initial testing with either no, one or two irrelevant attributes being incongruent found largely additive effects of each type of conflict in the Stroopon and a sub-additive effect in the Flankon. Hence, the final experiments focused on the Stroopon where the gain in conflict-effect magnitude was large, and included only trials with double congruence or double incongruence to maximize the number of responses collected per condition.

In this work we fine-tuned these modifications over three preliminary experiments, each with many participants performing a small number of trials. A final experiment focused on the best versions of the tasks, with participants performing many more trials to test the increased reliability predicted by Eqs. 1, 2. Contra this prediction, a survey of Stroop studies found no effect of trial numbers on reliability[36]. This might occur because of greater measurement noise due to fatigue and disengagement, increasing the trait-precision denominator, or because task automation due to practice[37] reduced individual differences in task performance, reducing the numerator. Overall, we predicted that the multiple sources of increased complexity and attempts to increase the conflict effect would lead to improved reliability. On the suggestion of reviewers, we conducted non-gamified versions (Fig. 3) of the final experiment, while keeping the initial refinements, to test the impact of gamification. Overall we

### Table 1 | Tasks used in each experiment

| Experiment | Flanker | | Simon | | Stroop | | Stroopon | | Flankon | |
|---|---|---|---|---|---|---|---|---|---|---|
| | 1 | 2 | 1 | 2 | 1 | 2 | 1 | 2 | 1 | 2 |
| Initial | Yes | Yes | Yes | Yes | X | X | Yes | Yes | Yes | Yes |
| First | Yes | Yes | Yes | Yes | X | X | Yes | Yes | Yes | Yes |
| Second | X | Yes | X | Yes | X | X | X | Yes | X | X |
| Final—gamified | Yes | Yes | X | Yes | X | Yes | Yes | Yes | X | X |
| Final—non-gamified | X | Yes | X | X | X | X | X | Yes | X | X |

1 = single response, 2 = second response.

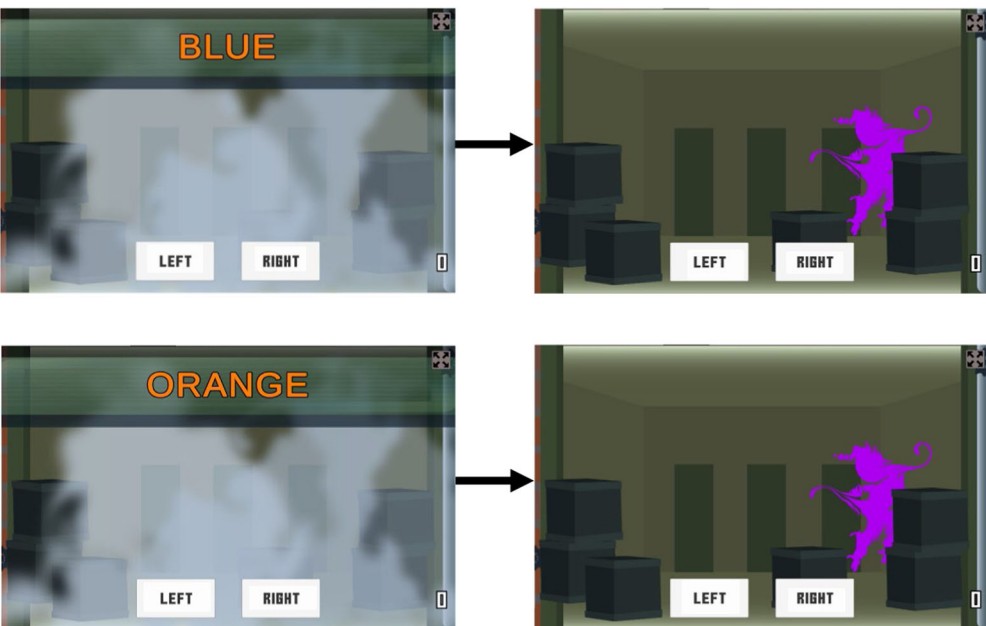

**Fig. 2 | Illustrations of the Stroop task with double-shot.** Stimuli are colored words presented in the middle of the display and a response is based on the color the text is written in. The top example shows an incongruent trial where the correct response is orange (or right, if orange = respond right and blue = respond left). The purple shield indicates a second response based on the word (i.e., blue). The bottom panel shows a congruent trial where the correct response is orange, followed by second response to the written word (i.e., orange).

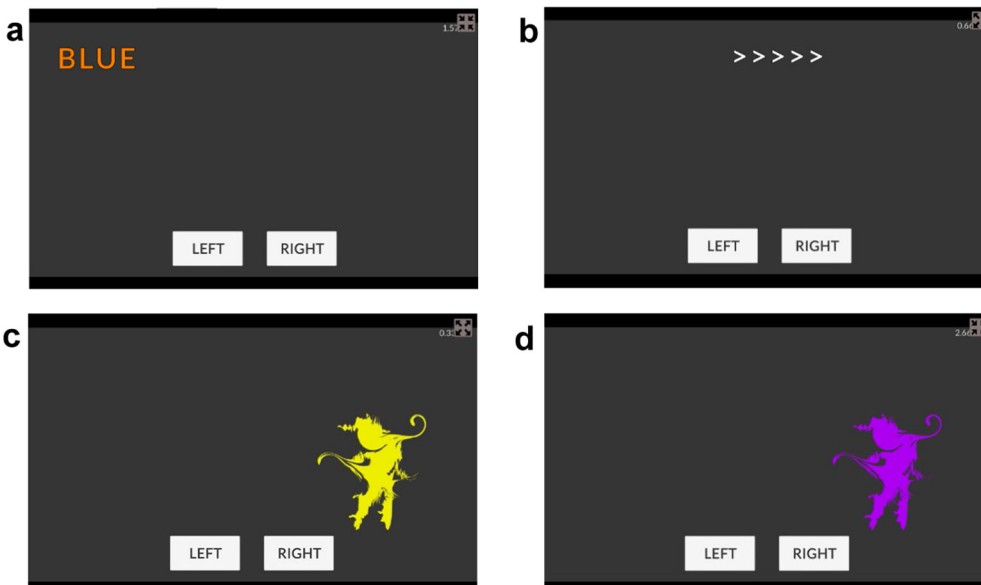

**Fig. 3 | Illustrations of the non-gamified tasks used in the final experiment.** Stroopon stimuli are colored words presented to left or right of display, the relevant information is stimulus color (i.e., orange): (**a**) double incongruent Stroopon trial (assuming blue = respond left; orange = respond right). Flanker stimuli are plain-colored arrows with the relevant information being the central arrow: (**b**) congruent flanker trial (all arrows point same direction). (**c**) is an example of a second shot Stroopon trial where the correct response to the location of (**a**) is left. (**d**) shows second response for (**b**) where the correct response to the flanking arrows is right.

found increased reliability in several of our tasks, including these non-gamified versions.

## Results

We report the findings for the final experiments, with the results of earlier experiments detailed in Supplementary Information. To establish a baseline for standard versions of the cognitive tasks, we analyzed archival Stroop, Simon, and Flanker data[17,38], as well as non-gamified versions of our tasks. In all cases we used hierarchical Bayesian analyses, extending upon previous approaches[21], to examine the effect of practice on reliability as trials accumulate. Participants completed multiple blocks of trials for a given task, thus, the analysis was applied to RTs for each trial from the first 4 blocks (48 trials in total), then the first 8 blocks (96 trials) and so on up to all 36 blocks (432 trials).

In each analysis, we fit linear mixed models using the BayesFactor package[39] to both raw RT and $\log_e$(RT-0.2). See previous work[40] for the appropriateness of the shifted Lognormal model of RT distributions implied by the transformation. Note also this analysis addresses how the magnitude of RT differences are usually proportional to overall RT[41]. Seconds units (s) are assumed throughout, and trials with RT < 0.25 s were removed from the analyses as anticipatory responses, so the logarithm was always defined. Details of pre-processing, code, and data used in the analyses are available at https://osf.io/5f8tz/. The transformation aimed to fulfill the normality assumption of the linear mixed models, which was critical because of our emphasis on estimates of variability. In Supplementary Figs. 1–3 we show that the transformation was effective in all cases, thus, we use it from here on.

For the first 4 blocks (i.e., 48 trials), we fit a model which assumed a fixed effect of conflict and random subject intercepts and slopes for the conflict effect (i.e., allowing for individual differences in its magnitude). For the remaining fits, we used a model assuming there was also an additive fixed effect of the 48-trial blocks (designated the standard model). Posterior samples from the models were used to obtain median estimates of the statistics displayed in Figs. 4–9 along with 95% credible intervals (i.e., the range in which according to the model the data occur with 95% probability[42]). To facilitate comparison, results are shown on the same scale, with the exception of the first column (Figs. 4, 6, 8), where the range is shifted up by around 1/3 of a second for our results, as responding was generally slower, and the scale for the number of trials required to reach a given reliability (Figs. 5, 7, 9), as fewer trials were required for our tasks.

Returning to the first column, untransformed RT for congruent and incongruent conditions are shown, with the dashed line calculated directly from RT and the gray band showing the model fit, which was good in all cases. Also provided are default Bayes Factors (BF) from the BayesFactor package comparing the standard model when applied to the full data set (432 trials) to a model (BF1) assuming no practice effect, and a model (BF2) assuming an interaction between practice and the conflict effect (analogous results for fits to fewer trials are provided in Supplementary Tables 1, 2). The BFs indicate how many times more likely the observed data are under the standard model than the comparison models[43], testing the adequacy of the assumptions about practice effects made by the standard model. Conventionally a ratio greater than 10 in favor of a model indicates strong support and greater than 100 indicates very strong support[44]. The BF1 results very strongly support speeding with practice in all but one case (Stroop task[38]) where the evidence is still supportive. The BF2 results very strongly support the conflict effect not diminishing with practice, with one exception (Flanker task[38]) where an interaction is ~7 (i.e., 1/0.14) times more likely than the standard model.

Given these minor exceptions, we use the standard model in all analyses to facilitate a uniform and simple interpretation. Note, however, that the additive practice effect model does not imply that the conflict effect on the natural (seconds) scale is unchanging with practice. Rather, as the conflict effect tested and displayed in the figures is on a log scale, it implies that the ratio of incongruent to congruent RT (with 0.2 s subtracted) is constant, implying a proportional decrease in the conflict effect measured in seconds due to the overall speeding caused by practice. Columns 2–4 (Figs. 4, 6, 8) display the results on the log scale in order to be consistent with the trait-precision results in column 5, but for ease of interpretation columns 2–4 also report their respective statistics for the full data set in seconds. Further, support for the standard model within levels of aggregation

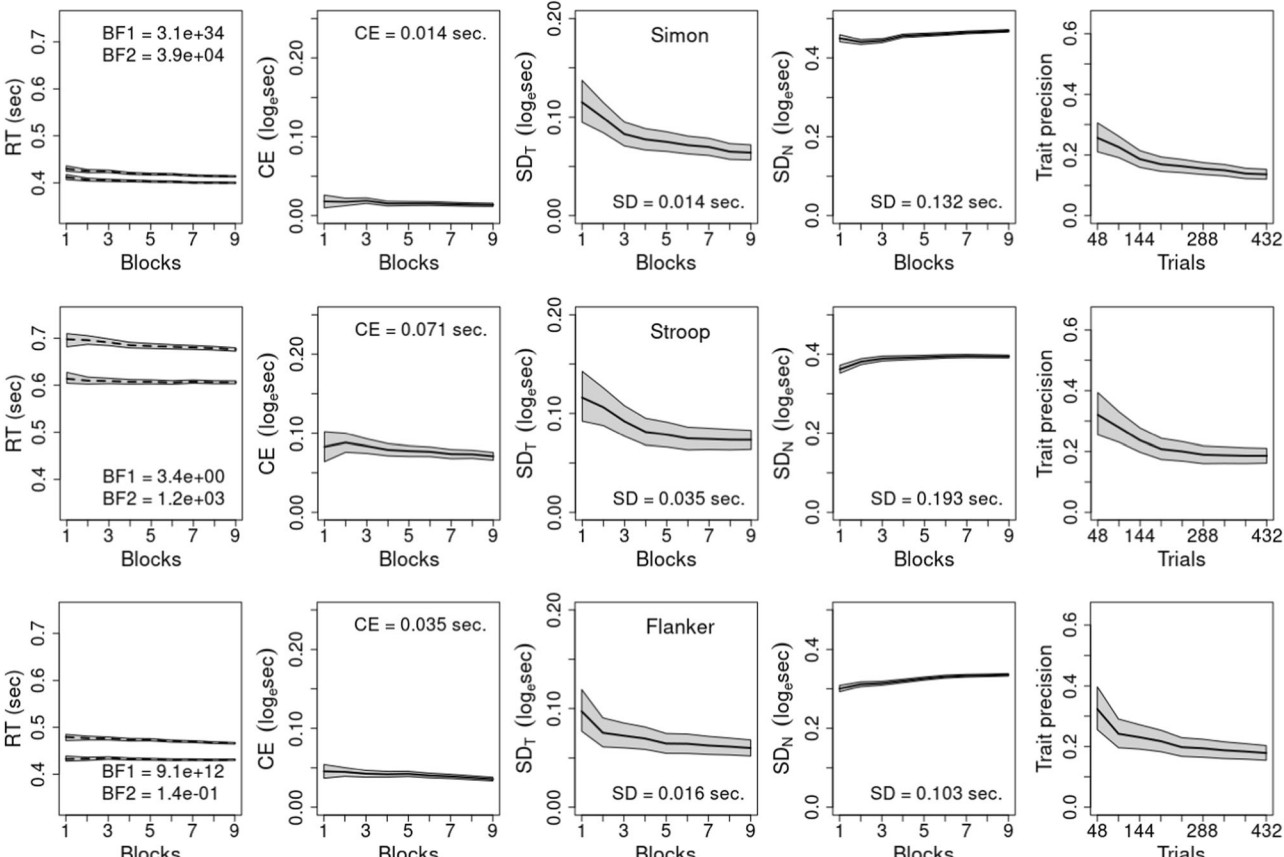

**Fig. 4 | Results for Hedge et al. data.** A model assuming additive practice (blocks) and conflict fixed effects, and random subject intercepts and conflict slopes. Blocks refer to cumulative sets of 48 trials and each row represents one task. In the first column, dashed lines show mean response time (RT) for congruent (lower line) and incongruent trials calculated directly from data and Bayes Factors (BF) compare the assumed model in the numerator to models with (1) no practice effect and (2) a practice × conflict interaction. In the remaining columns, solid lines represent median predictions from the model, and gray bands in all graphs show 95% credible intervals from fits of the assumed model. Values provided in columns 2–4 are seconds-scale equivalents for fits to all 432 trials for the conflict effect (CE) in column 2, and the trait and noise standard deviations (SD) in columns 3 and 4, respectively. Column 5 shows trait precision.

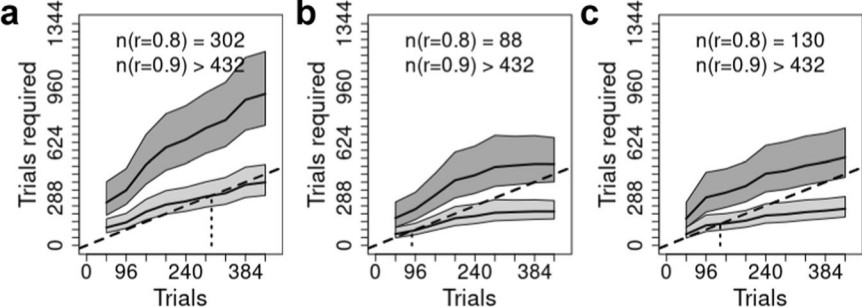

**Fig. 5 | Trials required for reliable results.** The total number of trials to achieve reliabilities (r) of 0.8 (lower line, light gray band) and 0.9 (upper line, dark gray band). Gray bands represent the posterior median 95% credible intervals. For every point on the x-axis we are predicting the number of trials required to achieve a reliability of 0.8 (light gray region) or 0.9 (dark gray region); the dotted vertical line descending from the dashed identity line (i.e., a line where x = y), and the values in text, provide the interpolated number of trials required for reliabilities of 0.8 and 0.9. Each panel represents a single task from Hedge et al.: (**a**) Simon, (**b**) Stroop, (**c**) Flanker.

does not necessarily imply that the log scale conflict effect remains unchanged as more trials are aggregated. Although Figs. 4, 6, 8 show that this is generally the case, there are some clear exceptions, such as the gamified Stroop2 and Flanker2 tasks.

Column 2 shows conflict effects are generally bigger in our tasks than those of Hedge et al.[17,38], while trait variability is generally greater and noise variability is generally smaller. As a result, trait precision shown in the final column is greater in our tasks, being greatest in the

Flanker tasks, second in the Stroopon tasks and least in the Stroop2 and Simon2 tasks. In all cases, precision decreases as more trials are aggregated, principally due to a decrease in trait variability (column 3). In our tasks, except for Stroopon, the attendant decrease in precision is slightly ameliorated by a concomitant decrease in noise (column 4). For the Stroopon, and archival tasks, noise increases and hence the decrease in precision is exacerbated. Note that measurement noise on the log scale is multiplicative on the natural scale, hence here, as is

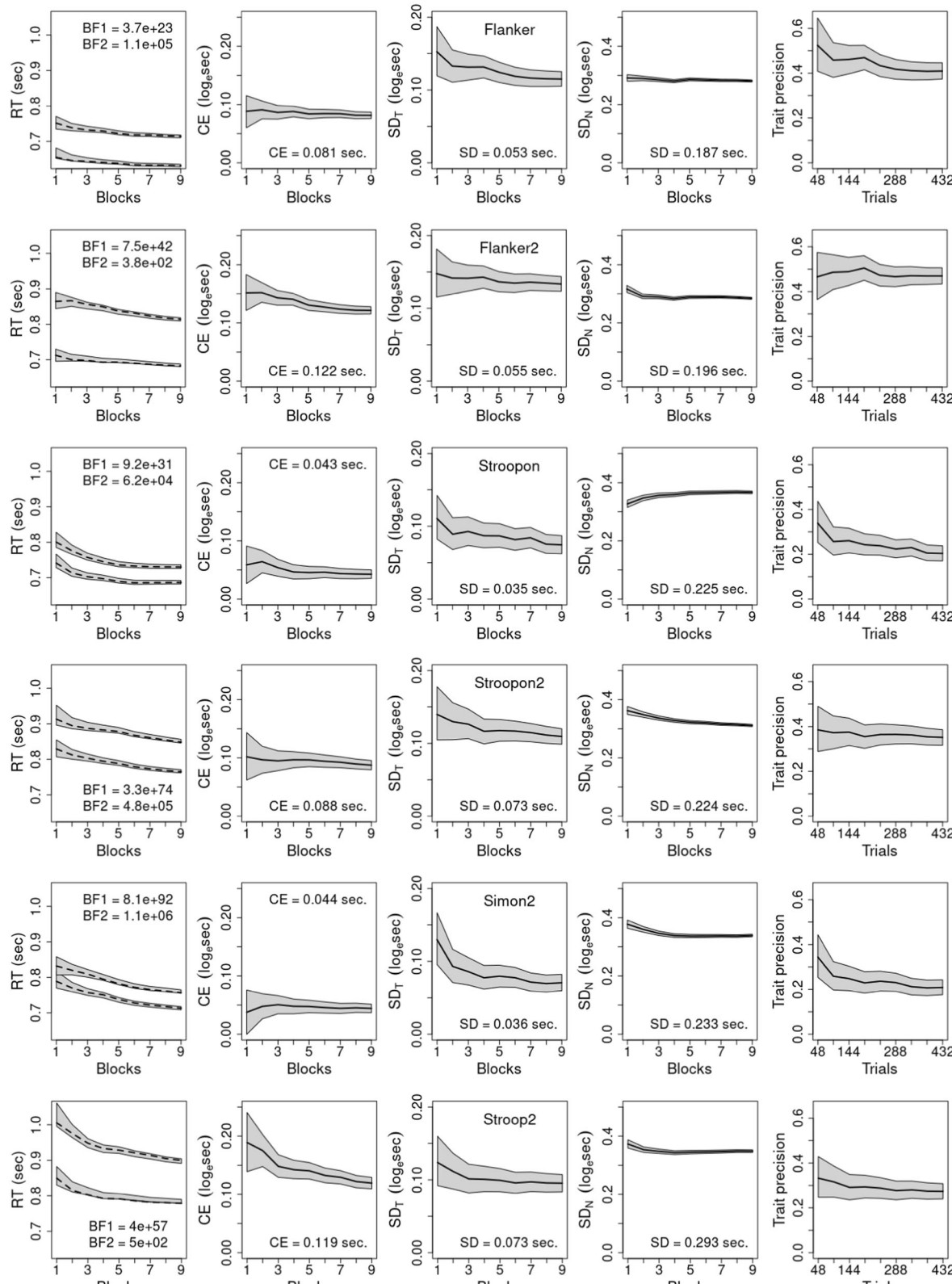

**Fig. 6 | Results for gamified experiment.** The model assumes additive practice (blocks) and conflict fixed effects, and random subject intercepts and conflict slopes. Blocks refer to cumulative sets of 48 trials and each row represents one task. In the first column, dashed lines show mean response time (RT) for congruent and incongruent conditions calculated directly from data and Bayes Factors (BF) compare the assumed model in the numerator to models with (1) no practice effect and (2) a practice × conflict interaction. In the remaining columns, solid lines reflect median results from the model, and gray bands in all graphs show 95% credible intervals from fits of the assumed model. Values provided in columns 2–4 are seconds-scale equivalents for fits to all 432 trials for the conflict effect (CE) in column 2, and the trait and noise standard deviations (SD) in columns 3 and 4, respectively. Column 5 shows trait precision.

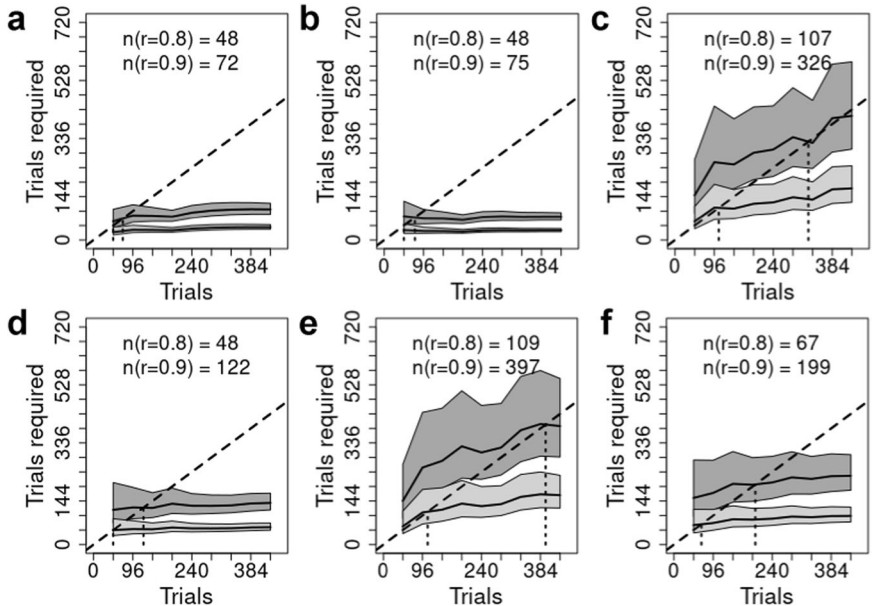

**Fig. 7 | Trials required for reliable results.** The total number of trials to achieve reliabilities (*r*) of 0.8 (lower line, light gray band) and 0.9 (upper line, dark gray band). Gray bands represent the posterior median 95% credible intervals. For every point on the x-axis we are predicting the number of trials required to achieve a reliability of 0.8 (light gray region) or 0.9 (dark gray region); the dotted vertical line descending from the dashed identity line (i.e., a line where x = y), and the values in text, provide the interpolated number of trials required for reliabilities of 0.8 and 0.9. Each panel represents a single gamified task: (**a**) Flanker, (**b**) Flanker2, (**c**) Stroopon, (**d**) Stroopon2, (**e**) Simon2, (**f**) Stroop2.

widely observed, the standard deviation of RT generally decreases with practice[25].

Summarized in Figs. 5, 7, 9 are the implications of our results regarding the trials required to achieve what is generally regarded as adequate (*r* = 0.8) and good (*r* = 0.9) reliability, which correspond to the standard deviation of individual differences in the conflict-effect trait being approximately 1.35 and 2 times the standard error of the conflict effect. In the archival tasks, precision is so poor that good reliability is not achieved even when all trials are aggregated. Adequate reliability is achieved at relatively modest trial numbers for Stroop and Flanker tasks, but the Simon requires over 300 trials. In contrast, our tasks achieve good reliability within the measured range. Analysis of the full set of over 800 trials reported for the archival Stroop and Flanker tasks[17] found good reliability was achieved at 520 and 614 trials, respectively. In comparison, the current Flanker tasks are best, with Flanker2 displaying a slightly weaker increase in the number of trials needed as aggregation increases. The Stroopon2 is second best, and contrasts with the Stroopon, which displays a much larger increase in the number of trials required. The Stroop2 task also performs well, but has a stronger increase with aggregation than the Stroopon2. Finally, Simon2 performs worst, although still much better than the Simon task in the archival data. Our gamified and non-gamified versions produced similar outcomes, with Flanker2 reaching a reliability of 0.9 in fewer trials in the non-gamified version, but the opposite being true for Stroopon2, where the gamified test is superior.

## Discussion

Our work joins a growing effort to improve the reliability of tests of cognitive control. Rather than abandoning conflict-task RT difference measures[28], we attempted to improve their reliability. We make irrelevant attributes harder to ignore by increasing their salience and similarity to relevant attributes, and by occasionally requiring a second response based on the irrelevant attributes. The latter requirement, along with gamification, also aimed to increase task complexity, and hence cognitive load, as well as maintaining good performance over longer testing sessions by increasing engagement. Finally, we combined different types of tasks, aimed at producing a larger combined conflict effect. Based on extensive pilot studies, we selected six tasks with various combinations of these features and assessed the number of trials necessary to obtain adequate (*r* = 0.8) and good (*r* = 0.9) reliabilities. Hierarchical Bayesian analysis[21] allowed us to directly estimate the factors determining reliability (*r*), trait, or true score, standard deviation, $\sigma_T$, and the standard deviation of trial-to-trial measurement noise, $\sigma_N$. As a comparison, we also applied this analysis to standard conflict-task data[17,38].

In all tasks, more trials strongly increased reliabilities, but there were diminishing returns, primarily because individual differences also decreased. The reliability of these tasks was increased relative to archival data, principally due to both increased conflict-effect magnitudes and individual differences. In the gamified versions, for the Flanker task, this resulted in a reduction in the number of trials required to reach adequate and good reliability by factors of 2.7 and 8.5, requiring 48 (or less) and 72 trials in total, respectively. The Stroopon2 task also performed well (requiring 48 or less and 122 trials, respectively). However, this was only when, on a randomly selected 1/3 of trials, participants made a second response based on either the word or its location (Stroopon2), after making a first response based on stimulus color. The Stroop and Simon tasks with the second response requirement were less reliable, requiring 67 and 199 trials, and 109 and 397 trials, respectively, although this was better than the standard tasks[17,38]. The non-gamified versions performed similarly: adequate and good reliability were reached in 48 and 53 trials for Flanker2, and 49 and 187 trials for Stroopon2. Once again, the Flanker task slightly outperformed the Stroopon. In summary, these results indicated that the Flanker and Stroopon2 tasks achieved good reliability in a number of trials that meets the practical requirements of studies using large test batteries.

What were the causes of these improvements? First, gamification was clearly not necessary. Perhaps our participants were sufficiently motivated to maintain their engagement without gamification. It also suggests that any increase in task complexity attendant to gamification was not essential. Being able to use simple, non-gamified tasks that are typical in experimental investigations for increased control, and hence the ability to isolate the causes of any improvement, are important.

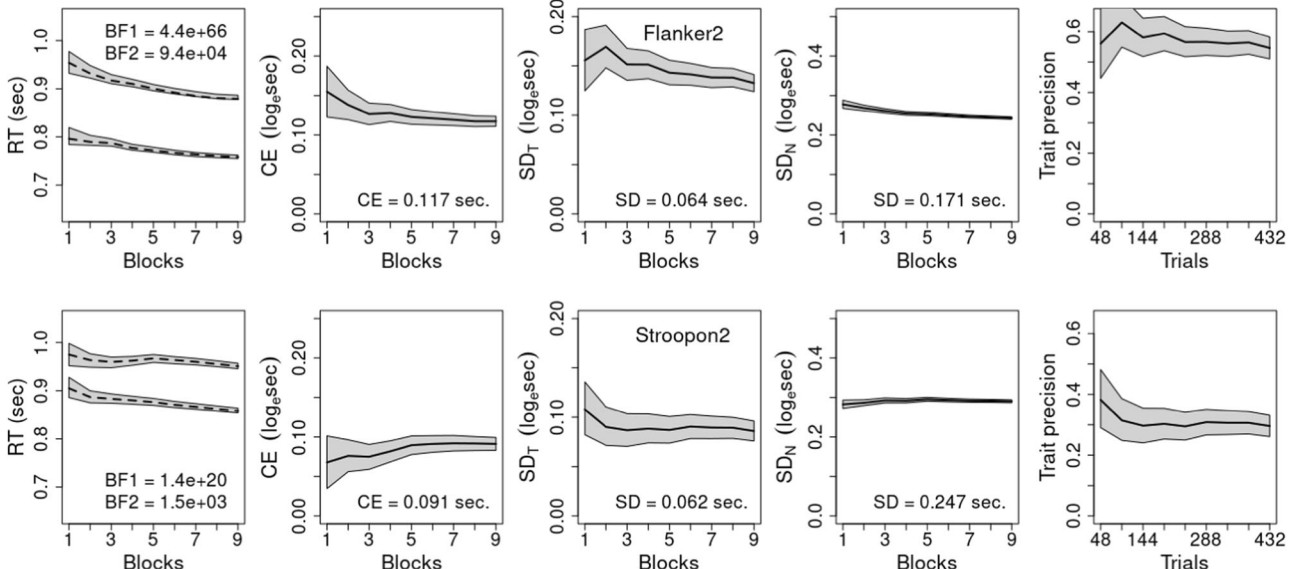

**Fig. 8 | Results for non-gamified experiment.** The model assumes additive practice (blocks) and conflict fixed effects, and random subject intercepts and conflict slopes. Blocks refer to cumulative sets of 48 trials and each row represents one task. In the first column, dashed lines show mean response time (RT) for congruent and incongruent conditions calculated directly from data and Bayes Factors (BF) compare the assumed model in the numerator to models with (1) no practice effect and (2) a practice × conflict interaction. In the remaining columns, solid lines reflect median results from the model, and gray bands in all graphs show 95% credible intervals from fits of the assumed model. Values provided in columns 2–4 are seconds-scale equivalents for fits to all 432 trials for the conflict effect (CE) in column 2, and the trait and noise standard deviations (SD) in columns 3 and 4, respectively. Column 5 shows trait precision.

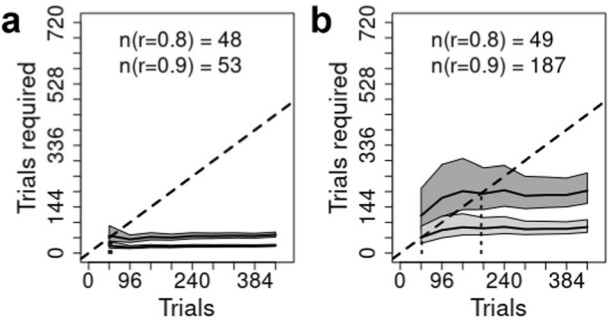

**Fig. 9 | Trials required for reliable results.** The total number of trials to achieve reliabilities (r) of 0.8 (lower line, light gray band) and 0.9 (upper line, dark gray band). Gray bands represent the posterior median 95% credible intervals. For every point on the x-axis we are predicting the number of trials required to achieve a reliability of 0.8 (light gray region) or 0.9 (dark gray region); the dotted vertical line descending from the dashed identity line (i.e., a line where x = y), and the values in text, provide the interpolated number of trials required for reliabilities of 0.8 and 0.9. Each panel represents a single non-gamified task: (**a**) Flanker2, (**b**) Stroopon2.

On the other hand, gamification was not deleterious, so it could be useful in circumstances or populations where engagement would otherwise be inadequate. It may also be that the task pacing used in both versions (short sets of 12 trials with breaks in between) was beneficial for engagement.

The key drivers of the reliability in our Flanker tasks are likely related to the signal-detection stage of processing. Irrelevant attributes were large, salient, and located very close to the relevant (target) attribute, and location uncertainty made it difficult for participants to quickly exclude interfering information through spatially-based selection[45]. Many different types of Flanker displays have been shown to produce robust experimental effects[6], but we are not aware of any systematic investigation of effects of display types on reliability. A reviewer also suggested for our incongruent displays, close spacing meant adjacent arrows formed a unified diamond shape (i.e., the target and distractor to its right and left in >>><>>

and <<><< displays, respectively) to pop-out and then require further processing to separate into its constituents. This hypothesis is consistent with the low reliability produced by Hedge et al.'s task[17], where the arrows had weaker grouping as they were arrayed vertically, but clearly further research is required to determine the exact roles played by such potential moderators and what components of reliability they affect (i.e., effect magnitudes, individual variability, and/or measurement noise).

For Stroopon2, likely causes of improvement are the increased conflict effect from combining Stroop and Simon interference, and the second response ensuring that participants encode irrelevant information. The latter factor appears to have primarily acted by increasing the conflict-effect magnitude, and weakening deleterious effects of changes with practice, perhaps due to discouraging the adoption of strategies, such as blurring vision to reduce Stroop interference. Thus, if the second-response requirement increased task complexity in a way that lessened participants capacity to combat interference, then it indicates the factors mediating the increase, such as having to hold two response rules in working memory, are different from those related to gamification.

One might question whether these potential causes of increased reliability also reduce the validity and/or specificity of the constructs being measured. We argue that the factors related to encoding do not decrease validity if interest focuses on the cognitive control of interference, but may do so if interest focuses on control methods that reduce the encoding of interfering information. Previous investigations that have combined the Simon task with either the Stroop or Flanker tasks[46–52] have focused on whether interference occurs in the same or different processing stages. Most have concluded, based on additive-factors logic that they address different stages. However, recently doubt has been cast[53] on this conclusion, suggesting that Stroop and Simon effects index overlapping types of conflict. Consistent with this, recent research[54] reports support for a shared mechanism for interference control across spatial Stroop and Simon tasks, but not a Flanker task, in an exploratory factor analysis. The authors suggest this is because interference control in the Flanker task is mediated by facilitation of target information, whereas in the other

tasks it is mediated by an inhibitory mechanism. It may also be that for the Flanker task, control is applied at an earlier signal-detection stage, whereas it occurs at a later stage for the other tasks. The fact a second-response was beneficial to the Stroopon but not Flanker, further suggests different types of cognitive control. Although it might be that strategically reducing the encoding of irrelevant attributes is easier to achieve, and so more likely to occur without a second-response requirement, in the Stroopon task. In any case, in light of these considerations and recent findings, it is fortuitous in terms of minimizing any loss of specificity that we found the combination of Stroop and Simon tasks was most beneficial. Again, further research is required to develop a more detailed understanding. However, pragmatically, task combinations provide a promising way forward given the lack of reliability otherwise found.

In terms of limitations, we note our reliability estimates correspond most closely to traditional split-half techniques, and test-retest reliability will likely be lower given that the longer time scales involved enable greater state variation. A reviewer pointed out that the stability of individual differences at longer time scales is typically of greater importance, and so cautioned that our usage of trait to describe characteristics that are stable at shorter time scales may be misleading. We retained the term because we see trait stability as falling along a continuum, thus one might raise similar concerns with respect to stability across days, weeks, months and so on. However, we agree that for applications relying on longer-term trait stability it will be important to perform test-retest analyses at a corresponding time scale (using appropriate hierarchical Bayesian analysis methods[55]). Further, our Mechanical Turk (MTurk; https://www.mturk.com/) sample collected during 2021 is a challenge to the comparability of participant populations on which our results and those of previous investigations are based, particularly given evidence that MTurk has become more diverse during the COVID-19 pandemic[56]. Work quantifying how reliability varies across different demographics seems warranted.

Our analysis deals with reliability, and not the concept of validity. Increasing reliability does not necessarily affect validity, but it is key to correlational approaches that attempt to establish the validity of domain-general control mechanisms[28], as low reliability leads to low measured correlations even when a common mechanism is present. The evidence establishing the validity of the tasks we have relied upon identifies mechanistic explanations and is experimental in nature, and so low reliability does not bring it in to question. In contrast, fundamental problems have been identified with the practice of attempting to establish validity through associations among individual-difference attributes both in general[57], and specifically with respect to cognitive control[31], leading to calls for it to be abandoned. Hence, we believe it important that any inferences about general control mechanisms enabled by the increased reliability of our tasks be complemented by evidence about explicit psychological and neural mechanisms through which they could plausibly be explained. Where these tasks are to be used in specific applied domains (e.g., predicting impulse control in substance use) traditional correlation-based approaches remain useful.

Our analyses are limited in focusing only on RT differences, which forgo potential information contained in accuracy differences and may be subject to confounding from speed-accuracy tradeoffs, which can dissociate effects on RT and accuracy differences. Fortunately, RT and accuracy differences are often strongly positively correlated when taken over conditions that are randomly intermixed, as was the case for our tasks, but not when they are blocked[38]. In the latter case the ability of evidence accumulation models[58] to partial out the effects of speed-accuracy tradeoffs makes them attractive. Even when conditions are mixed, their ability to combine information for RT and accuracy to measure conflict effects regarding the rate of evidence accumulation is potentially advantageous. Given our mixed task design, the increased reliability of RT differences we obtained seems likely to translate to increased reliability for rate differences, but further research is required to explore this possibility.

In conclusion, the tasks which we developed build on the long history of using RT difference measures of decision-conflict in the experimental literature while requiring only a modest number of trials to attain the reliability required for applications in differential psychology. Our tasks are freely available (https://osf.io/5f8tz/) in the hopes that they will be useful in domains ranging from theoretical differential psychology research to providing training and personnel selection guidance in the many high-stakes applications where the resolution of decision-conflict is a key skill. For future research on different tasks using RT difference measures, we recommend our analytic approach based on the methodology in[12], and careful consideration of the effects of practice when increasing trial numbers, to guide and rigorously evaluate development.

## Methods

In total we conducted three initial experiments as well as a final experiment (including non-gamified versions of two tasks). Participants were recruited on the MTurk platform, which directed them to the PlayUR (https://playur.io) experiment management system, which enables participants to play games developed in Unity (https://unity.com) through a web browser. All participants were aged over 18 years old and no other demographic information was collected. The six task variants in the final gamified experiment were selected from a larger set based on the three preliminary experiments, which we briefly summarize below. A total of 1027 participants each performed a relatively small number of trials (no participant as identified by their PlayUR username participated in more than one experiment). Details of the initial experiment are reported in Wells et al.[59] and in Supplementary Information for the other preliminary experiments (designated Experiments 1 and 2 here).

### Task overview

The tasks themselves remained identical over all experiments, with a few exceptions noted below. From Experiment 1 onward, after participants completed a tutorial on their assigned task, they had eight chances to correctly complete four trials of that task; if they did not pass this test they could not continue to the main experiment.

Several gaming mechanics were employed that have been shown to improve competence, satisfaction, and task meaningfulness[60]. The tasks were embedded in a room-clearing narrative, where participants had to enter a room and use the information in a display at the top of the screen to determine if an enemy was hiding on the left or right and press a corresponding button as quickly as possible. Auditory feedback indicated accuracy and speed, with points given for a correct response and increased speed relative to the last trial. The tasks were organized into a series of 12-trial blocks. A points total was continually visible and accrued over blocks. After each block, accuracy feedback was supplied.

### Preliminary experiments

In the initial experiment, two groups of 72 participants performed 24 trials in each of the single and double-shot (a second response was required on a random 1/3 of trials) versions of their assigned condition. These being either the Simon and Stroopon (i.e., a written color as in the Stroop presented on the left or right of the screen), or the Flanker and Flankon (i.e., combined Flanker and Simon interference, with responses required to Flanker displays presented on the left or right of the screen) tasks.

In the Simon task, a blue or orange rectangle was presented on the left or right of the screen, and displays could be either congruent (e.g., if blue = left response, then the rectangle is presented on the left) or incongruent (e.g., blue rectangle on the right). In the Flanker task, the stimulus (e.g., >>>>>) was presented near the middle of the screen with

the central target arrow's location being, with equal probability on each trial, in the exact center or to the left or right by one- or two-character widths. This spatial uncertainty was introduced so that participants could not focus on the target arrow location with any certainty before the trial began. Again, the display was either congruent (all arrows point in the same direction) or incongruent (middle arrow points opposite direction to the flanking arrows). The Flankon and Stroopon tasks consisted of four conflict conditions, double incongruent (e.g., ORANGE written in blue, requiring a left response, presented on the right of the screen), single incongruent (either in location, e.g., BLUE written in blue, requiring a left response, presented to the right or word, e.g., ORANGE written in blue, requiring a left response, presented to the left) and double congruent (e.g., BLUE written in blue, requiring a left response, presented on the left of the screen).

The pre-registered sampling plan (https://osf.io/y4sbh) for the initial experiment used a sequential Bayes factor method[61], with a minimum sample size of 72 in each set and a maximal sample size of 216 comparing single vs. double-shot conflict effects in the Simon or Flanker tasks. Data collection halted at the minimal sample size based on evidence favoring no difference between conflict effects in single and double-shot conditions.

In Experiments 1 and 2, each participant performed 48 trials (12 trials × 4 blocks) in only one task so that carryover effects on differences between tasks could be ruled out. The basis of our sampling plan also shifted to collecting enough participants to obtain reasonably precise standard errors on effect and reliability estimates. We set this at more than 70 participants per condition based on the results of the initial experiment. In most cases, we exceeded the required number due to the rapid nature of data collection on MTurk, and we decided to keep the full samples in all cases. This resulted in 670 participants in Experiment 1, across eight between-subject versions of the conditions in the initial experiment: Flanker ($n = 80$), Flanker2 ($n = 80$), Flankon ($n = 82$), Flankon2 ($n = 82$), Simon ($n = 85$), Simon2 ($n = 85$), Stroopon ($n = 88$), Stroopon2 ($n = 88$). Again, double shots occurred on a random 1/3 of trials for those conditions with this requirement (e.g., Flanker2, Flankon2). The results of the first two experiments indicated that the Flanker and Stroopon were the most promising tasks in terms of reliability, with the Flankon offering no improvement over the Flanker. We, therefore, dropped the Flankon from further experimentation. Experiment 1 found some evidence for increased effect sizes in double over single-shot conditions, but no appreciable increase in reliability.

Involving 213 participants, Experiment 2 required second responses on all trials of the Flanker ($n = 70$), Simon ($n = 73$), and Stroopon ($n = 70$) tasks. As there was no evidence of improved reliability over the version with the second response on 1/3 of trials, which had the advantage of being quicker to perform, we adopted the 1/3 version in the final experiment. Although the double-shot manipulation did not have a marked effect in the preliminary experiments, we continued to include double-shot conditions because we hypothesized that they may discourage participants from discovering and adopting strategies (i.e., blurring the eyes in the Stroop task) to reduce the encoding of irrelevant attributes over the course of the longer final experiment.

**Final experiment—gamified.** The final experiment comprised six between-subjects conditions. Each condition consisted of one of the following tasks: single-shot Flanker or Stroopon, or double-shot versions of Flanker (Flanker2) or Stroopon (Stroopon2), as well as double-shot versions of the constituent Simon and Stroop tasks of the latter (Simon2 and Stroop2). The Stroopon task was modified to consist of half double incongruent and half double congruent stimuli (Fig. 1g, h). Each task consisted of 432 trials. Given this 9-fold increase in the number of trials, and the attendant increase in measurement precision at the participant level[62], but also the increased payment per participant, we reduced our sampling target to 30 participants per condition

(i.e., task). The large number of participants involved in the initial experiments, and the fact that we obtained similar findings in the final experiment, further serves to justify the reduction in participant numbers. Data quality is also enhanced by the inclusion of the non-gamified experiment.

## Participants
A total of 256 participants attempted the experiment; 61 failed the tutorial, and 9 had incomplete data. Another three were excluded for low accuracy (<60% overall), and one each for too many anticipatory responses (>10% of trials with RT < 0.1 s), and for non-responding (>10% of trials not completed within 4 s), resulting in a final sample of 181 participants (30 in each task, and 31 in Stroopon2). All participants received $1.00 USD for attempting the study. For those completing both sessions, an additional payment of $9.00 was given. The bonus that accrued throughout the experiment was up to $0.20 per block resulting in a total bonus of up to $7.20. Participants were required to have a human intelligence task (HIT; MTurk terminology for a task or study) approval rate of above 95% to be eligible to undertake the study. Informed consent was obtained from all participants and the research was approved by the University of Tasmania's Human Research Ethics Committee.

## Design and materials
The total duration of the experiment was approximately 65 min, with approximately 10 min for the tutorial. Participants were randomly allocated to one of the six experimental conditions and responded via the Z and / keys on their desktop or laptop computer. In the Flanker tasks, the right-hand / key always corresponded to right-facing arrows and the left-hand Z key to left-facing arrows. Otherwise, the two possible response-key assignments were counterbalanced across participants: for example, in the Simon task, Z for blue and / for orange stimuli or / for blue and Z for orange stimuli.

After a response on single-shot trials, the enemy appeared as a black shape, crosshairs appeared over the response location, and only the correct response key, now named to indicate the correct response, remained on the screen. For double-shot Flanker trials, the enemy appeared with a purple shield and a response based on the flanking arrows' direction was required. For double-shot trials in the Simon task, a yellow shield required a response based on the stimulus location. In the Stroopon task, a yellow shield required a response based on the stimulus position and a purple shield a response based on the word (Fig. 1i, j). In the Stroop task, double-shot trials displayed a purple shield and required a response based on the word (see Fig. 2).

## Procedure
To be eligible to begin the study, participants had to check a box confirming they have normal, or corrected-to-normal, vision and successfully completed two multiple choice English proficiency questions consisting of a sentence missing one word (e.g., He really didn't _____ to interrupt her. Please choose only one of the following: (a) hope, (b) gain, (c) mean, (d) suppose).

A tutorial then guided participants through each component of the task pertaining to the condition they were randomly assigned to. Each component was introduced with prompts explaining the aim of the task and how to respond (e.g., respond according to the direction of the central arrow). Next, they completed the component unprompted. The prompted and unprompted blocks each consisted of four trials (including both incongruent and congruent stimuli). An error message was displayed when an incorrect response was given, and the response had to be corrected to continue. For example, in Flanker2, the standard Flanker task (with then without prompts) was introduced, then Flanker with double shots (with then without prompts) was presented. Participants received points for correct responses and the total was constantly displayed. There were no time

limits imposed on trials, however, a "Hurry up!!" message appeared when participants responded slowly (i.e., >2 s).

Participants then received eight chances to successfully complete four unprompted trials of each component (e.g., Flanker and Flanker2) without error in any element, before progressing to the experiment. Again, errors were required to be corrected to advance. Where eight attempts had been made without successful completion, participants were asked to exit the study and paid for their time. Once all trials were correct in a single attempt, participants proceeded to the experiment without the need to complete any remaining attempts.

The main experiment was separated into two sessions, the first consisted of 16 blocks and the second 20 blocks. The sessions were divided by a compulsory 10-minute break; however, participants were free to break for longer if desired. The bonus accumulated throughout the experiment and was visible to participants at the end of each block. This included the amount they earned in the preceding block as well as a running total. At the conclusion of both sessions, participants were informed of the total bonus they had accumulated and were given a completion code for MTurk.

**Final experiment—non-gamified.** The non-gamified experiment consisted of two between-subjects conditions: Flanker2 and Stroopon2. In order to match the gamified version, the final sample consisted of 60 participants; 30 in each condition.

### Participants
Of a total 81 participants who attempted the experiment, 14 failed the tutorial, 6 had incomplete data, and 1 was excluded for low accuracy (<60% overall). As for the gamified version, all participants received $1.00 USD for attempting the study. For those completing both sessions, an additional payment of $15.00 was given. This amount was roughly equivalent to the average bonus paid to participants in the gamified version. The bonus did not accumulate over blocks as we aimed to keep this version as close to a standard, non-gamified cognitive task as possible, in particular we tried to approximate the design of the Hedge et al.[17,38] studies. All other requirements were the same as the gamified version.

### Design and materials
The Design and Materials remained consistent with the gamified experiment, with the following exceptions. During the experimental blocks, trial feedback was not provided, however, overall accuracy and RT were given at the end of each block. To eliminate the gamified aspects, instead of an enemy appearing to indicate a double-shot trial, participants were told to respond based on the colored shape displayed. The color of these shapes was equivalent to the gamified version. The other crucial difference in the non-gamified study was that stimuli appeared on a dark gray background and there was no scoring system in place (see Fig. 3).

### Procedure
Eligibility remained consistent with the gamified study and participants did not complete any of our previous experiments. The tutorial was largely similar as well, although no points were awarded, and the display was plain aside from the stimuli presented. This was to remove any gamified characteristics that were otherwise present.

Again, the main experiment was separated by a mandatory 10-minute break into two sessions, with 16 and 20 blocks, respectively. Upon completion of the second session, participants were given a completion code for the HIT on MTurk. The bonus awarded was standardized across participants and was not dependent upon performance.

### Statistical analysis and data visualization
Data analysis and figure generation used R v3.6.0 and RStudio v1.1.383. The following packages were employed: BayesFactor v0.9.12-4.2, coda v0.19-3, Matrix v1.3-2v, stringr v1.4.0. Custom code was otherwise developed for data analysis.

### Reporting summary
Further information on research design is available in the Nature Portfolio Reporting Summary linked to this article.

### Data availability
The data generated in this study have been deposited on OSF: https://osf.io/5f8tz/. The Hedge et al.[17,38] data have been deposited on OSF: https://osf.io/cwzds/ (Stroop and Flanker tasks) and https://osf.io/btsrw/ (Simon Task). Access is also provided at the first OSF link.

### Code availability
All code (including custom code) to reproduce the analyses described herein is available through OSF: https://osf.io/5f8tz/.

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

## Acknowledgements

This research was funded by Australian Army Headquarters (Land Capability Division) and an Australian Research Council Discovery Grant to A.H., J.S., and M.P. (DP200100655). A.H. was supported by a Révész Visiting Professor Fellowship from the University of Amsterdam. D.M. was supported by a Vidi grant (VI.Vidi.191.091) from the Dutch Research Council. T.K. was supported by an Australian Government Research Training Program Scholarship from the University of Tasmania.

## Author contributions

A.H. and E.A. conceptualized the study. A.H. designed the tasks. L.W. programmed the experimental tasks under the guidance of T.K., A.H., and I.L. T.K. and A.K. managed data collection. A.H. and T.K. performed the data analysis. A.H., T.K., and D.M. drafted the paper and all other authors (L.W., I.L., K.D.S., A.K., M.P., J.S., and E.A.) provided feedback.

## Competing interests

The authors declare no competing interests.
