## [Peer Review File · Nature Communications]

Calibration of cognitive tests to address the reliability paradox for decision-conflict tasksREVIEWER COMMENTS

Reviewer #1 (Remarks to the Author):

In the present manuscript, the authors propose a solution to the reliability paradox – the general finding that tasks which produce robust and reliable group effects tend to be unreliable in differential research). Their solution entails administering gamified versions of popular cognitive conflict tasks (Stroop, flanker, and Simon, which are known to be good examples of the reliability paradox), including combining Simon tasks with the other two tasks, designed to obtain larger RT interference effects and therefore increased reliability. Using hierarchical analysis, the authors found that these gamified and combined measures generally demonstrate higher reliability, some of which in under 100 trials.

This manuscript has a number of strengths, including that it concerns an important topic in individual differences and is well-written. There have been many studies using various methods over the last several years trying to find solutions to the reliability paradox and most of them, frankly, fail to show significant improvements. I also highly appreciate that the authors made their tasks available to other researchers and encouraged their use.

That said, this manuscript is quite methodological and psychometric in nature, and so I believe it would more appropriate for an experimental or methodological journal than Nature Communications. I also believe there are some limitations of the current research, and important discussion of these (which would likely increase the impact of this paper considerably) is rendered impossible given the word requirements of Nature Communications.

Given these concerns, my recommendation would be to reject this manuscript. However, I think there are merits to this study and that it does advance the field of science. I think it would be a much better fit in a different journal – one that would allow more space to give the methods and results more discussion than is currently possible. In short, I think this manuscript is not well suited for publication in a multi-disciplinary with strong length requirements. Below is a list of my comments and concerns.

Major comments and concerns:

1. As the authors note on p. 10 lls 233-234, this work does not address the question as to whether validity is improved when using these more reliable measures. This is a major limitation. I am very skeptical that merely improving the reliability of these tasks would result in appreciable gains in terms of validity, especially when sticking with RT contrasts.

Improving the reliability of conflict tasks is an important step, but there are quite a few ways this can be done in a way that is not theoretically interesting in terms of assessing the underlying mechanisms and cognitive processes believed to be involved here. For example, Hedge et al. (2021) demonstrated that conflict tasks have very little validity for assessing cognitive conflict. This is not only a reliability problem, but, as they noted, even if reliability was improved and strong correlations were found among conflict tasks and various other measures, they would likely be attributable to factors such as response cautiousness and/or processing speed (i.e., not attention, inhibition, or conflict resolution). Without a proper validation of these new measures, I would assume something similar here. Studies over and over have shown that even with (somewhat) reliable versions of conflict tasks, there is still little-to-no correlation with other executive functioning tasks when using difference scores, suggesting that the problem with these measures is not only that difference scores are involved in calculation of performance (see Draheim et al., 2019; also see Rey-Mermet et al., 2019 who noted that difference scores appear to not be as process pure as advertised). This is also related to the argument from Rouder and colleagues that, in short, these tasks simply do not appear to measure what they are thought to from the correlational perspective. Without any sense of validity (including criterion validity and discriminant validity), it is not possible to know whether the observed improvements in reliability are theoretically meaningful.

2. Related to the above point; P. 4 lls 98-99 the authors state in regard to abandoning RT difference

scores that "...this potentially discards advantages of difference scores in terms of validity." This is something deserving of more discussion because it is central to the hypotheses and approach the authors used.

The authors are operating with the hypothesis that improving the reliability and effect sizes in these tasks will improve their validity. The authors need to justify their reason for sticking with RT difference scores to measure performance in these tasks, despite a large body of evidence showing RT differences are, not just often unreliable, but display poor criterion-related validity.

There is ongoing debate as to whether RT difference scores are as process pure as many believe, with researchers increasingly recognizing that the problems with difference scores (primarily in RT) and conflict tasks more generally are not isolated to unreliability but also validity (e.g., Draheim et al., 2019; Hedge et al., 2021; Rey-Mermet et al., 2019). Note in the developmental/aging literature, it has long been recognized that RT Stroop effects are artificially increased in individuals with slower processing speed.

A point I have made before is that one of the issues with the logic of difference scores in the context of conflict tasks is that there is evidence that even congruent trials require controlled processing, to an extent, especially when presented in mixed-blocks designs as is often the case (in the tasks switching literature, this is known as a mixing cost.) This is particularly true when congruent trials are not more frequent than incongruent trials.

As one example, see Figure 2 from Heitz & Engle (2007) using the Eriksen flanker task (attached to this review). Notice how high vs. low working memory capacity individuals differ in their RT on compatible (congruent) trials. Using difference scores removes this source of individual variation.

Other studies have also found evidence of controlled processing in the baseline trials of various attention measures – including prosaccade trials of antisaccade tasks (Unsworth et al., 2004) and congruent trials in color Stroop (Kane & Engle, 2003).

3. The research presented in this manuscript is highly methodological in nature, using advanced statistics many researchers are not familiar with. There are also multiple experiments, multiple unique tasks and task combinations, comparisons to previous research, etc. The complexities of the study are difficult, if not impossible, to properly convey to the reader in a journal with such limited page space. I submit that it would be beneficial to both the authors and the reader to have a detailed introduction and explanation of methodology and results (including many more tables and/or figures), which is not possible in Nature Communications.

To this point, I found it difficult to evaluate the results because it was boiled down to just a couple pages and figures, and without mention of individual reliability estimates, correlations between tasks (if some of the same tasks were administered to the same participants – was not entirely clear), and so on. At points, I had to mostly take the authors' word regarding what they found, because I did not have access to a table or individual figures to interpret the results for myself. A longer manuscript could more thoroughly describe not just the results and the different experiments (with additional tables and figures for important aspects of the data omitted from the present manuscript, presumably due to length limits), but expand on practical recommendations, limitations, and other important factors which would allow the reader to better digest this research, and perhaps improve the impact of this line of work.

4. The comparison to results from Hedge et al. (2018) strikes me as not entirely appropriate, i.e., somewhat limited. It is useful to have a comparison to non-gamified versions of these tasks, but comparisons across labs and samples have limitations. For example, it is not possible to know whether non-gamified versions of these tasks given to the participants from the present experiments would also have showed larger effects and better reliability.

5. These data seem to be based on a very small group of subjects. The overall sample size of 181 participants for the final experiment is just barely enough for a validation study (i.e., enough to obtain stable correlation estimates; see Schönbrodt & Perugini, 2013), but the authors note on p. 14 that there were only 30 participants per condition. It is not entirely clear to me what a condition is in the context of this study (30 per task?). Either way, the authors need to make additional steps to justify how such a small sample size is appropriate for a validation study attempting to solve the reliability paradox.

Minor comments and concerns:

1. On p.3 ll 63, the authors state that the reliability paradox was recently identified by Hedge et al. (2018). This is not quite accurate.

The term "reliability paradox" was coined by Hedge et al. (2018) in a fantastic and now seminal paper, but the paradox had been identified long before Hedge et al., dating at least as far back as the 50s-70s with psychometricians such as Lee Cronbach and Fred Lord, among others, debating how to measure change. The paradox was also discussed by Overall and Woodward (1975) in a paper entitled "Unreliability of Difference Scores: A Paradox Measurement of Change", in which they noted that unreliability of difference scores is a concern for correlational research but argued that it was not a problem for experimental research because statistical power is actually increased at low reliability (and maximized at a reliability of 0), and by Logie et al. (1996) who noted that robust experimental effects do not translate to reliability at the individual level.

In regard to conflict tasks such as Flanker, Simon, and Stroop (and task-switching tasks), the reliability paradox was discussed (but not named) by Draheim et al. (2016), although the focus of that paper was more so the lower reliability of task-switching measures (and, again, this was based on previous research, I am by no means suggesting we identified the paradox). Friedman and Miyake (2004) also noted that difference scores were a likely cause of the low reliability and validity they found in measures of inhibition, including Stroop and flanker. There are a number of other papers that touch on the paradox by researchers (i.e., in the clinical or developmental space) who understand and appreciate the fundamental difference between experimental and differential research.

To summarize, while Hedge et al. named the "reliability paradox", it is not an entirely accurate characterization that they were the first to identify it, either generally or in regard to conflict tasks. The authors should consider modifying their introduction accordingly.

2.P. 5 116-119 the authors note that simply adding more trials to the task is likely not a great approach, and one reason being that fatigue or practice effects can influence reliability. This is a really great point. It might be worth mentioning a collaboration project (preprint) that counterintuitively found no correlation between number of trials and reliability of tasks using difference scores (analysis was based on a systematic review involving 46 studies, see p. 33 of von Bastian et al., 2020). Also important is that while Rouder et al. noted increasing trials can improve reliability, they also were skeptical of this approach being successful given it would take possibly as many as 1000 trials (or possibly more) to reach adequate reliability in some conflict tasks.

-Chris Draheim

References

Draheim, C., Hicks, K. L., & Engle, R. W. (2016). Combining reaction time and accuracy: The relationship between working memory capacity and task switching as a case example. *Perspectives on Psychological Science*, 11, 133-155.

Friedman, N. P., & Miyake, A. (2004). The relations among inhibition and interference control functions: a latent-variable analysis. *Journal of Experimental Psychology: General*, 133(1), 101-135.

Hedge, C., Powell, G., Bompas, A., & Sumner, P. (2021). Strategy and processing speed eclipse individual differences in control ability in conflict tasks. *Journal of Experimental Psychology: Learning, Memory, and Cognition*. Advance online publication. <http://dx.doi.org/10.1037/xlm0001028>

Hedge, C., Powell, G., & Sumner, P. (2018). The reliability paradox: Why robust cognitive tasks do not produce reliable individual differences. *Behavior Research Methods*, 50(3), 1166-1186.

Heitz, R. P., & Engle, R. W. (2007). Focusing the spotlight: Individual differences in visual attention control. *Journal of Experimental Psychology: General*, 136(2), 217-240.

Kane, M. J., & Engle, R. W. (2003). Working-memory capacity and the control of attention: the contributions of goal neglect, response competition, and task set to Stroop interference. *Journal of Experimental Psychology: General*, 132(1), 47-70.

Logie, R. H., Sala, S. D., Laiacona, M., Chalmers, P., & Wynn, V. (1996). Group aggregates and individual reliability: The case of verbal short-term memory. *Memory & Cognition*, 24(3), 305-321.

Overall, J. E., & Woodward, J. A. (1975). Unreliability of difference scores: A paradox for measurement of change. *Psychological Bulletin*, 82(1), 85-86.

Rey-Mermet, A., Gade, M., Souza, A. S., Von Bastian, C. C., & Oberauer, K. (2019). Is executive control related to working memory capacity and fluid intelligence?. *Journal of Experimental Psychology: General*, 148(8), 1335-1372.

Schönbrodt, F. D., & Perugini, M. (2013). At what sample size do correlations stabilize?. *Journal of Research in Personality*, 47(5), 609-612.

Unsworth, N., Schrock, J. C., & Engle, R. W. (2004). Working memory capacity and the antisaccade task: individual differences in voluntary saccade control. *Journal of Experimental Psychology: Learning, Memory, and Cognition*, 30(6), 1302-1321.

Von Bastian, C. C., Blais, C., Brewer, G. A., Gyurkovics, M., Hedge, C., Kalamala, P., Meier, M. E., Oberauer, K., Rey-Mermet, A., Rouder, J. N., Souza, A. S., Bartsch, A. S., Conway, A. R. A., Draheim, C., Engle, R. W., Fischkorn, G. T., Friedman, N. P., Gustavson, D. E., Koch, I., . . . Whitehead, P. S. (2020, July 27). Advancing the understanding of individual differences in attentional control: Theoretical, methodological, and analytical considerations. *PsyArXiv*. <https://doi.org/10.31234/osf.io/x3b9k>

Reviewer #2 (Remarks to the Author):

The paper addresses a topic that is certainly of wide interest – offering a solution to the difficulties of measuring individual differences in cognition (e.g. as evidence of cross-disciplinary interest, the paper whose data is used in figure 2, Hedge et al. 2018, has 700 citations, and that paper only highlighted and explained the problem rather than offering a solution).

The current paper makes the following valuable contributions:

- Shows that combining conflict tasks can provide more reliable individual differences in a feasible number of trials for many applications.

- Uses a more engaging gamified environment for the tasks, which likely contributes to the reliability by keeping participants on task for longer.
- Provides the code for these tasks openly and freely
- Deploys sophisticated analysis which, if I understand correctly, both helps the reliability of cognitive individual differences by handling practice effects and generally smoothing through the model fitting process, and also helps understand the nature of the data through separating 'trait' variance and measurement variance.

Thus overall, the paper puts the measurement of cognitive individual differences on a more positive footing, which is very much needed. It gives researchers something to work with, despite the limitations and suggestions listed below.

However, there are several ways I would suggest the paper can be improved if it is to be published in a general audience journal.

- The accessibility of the text will prove challenging for most readers in many sections.
- The accessibility of the figures also needs to be improved, even for specialist readers. Try to avoid acronyms as axis labels, because it makes a reader's job so much harder to translate them all. Most importantly, the most critical message of the paper is in column 6 of the figures, but it is extremely difficult to understand what these plots are plotting and the rationale for it. The y axis of $n=2xL$ is not explained in the main text or in the figure legend, and nor is why the critical number of trials is where trials vs $n=2xL$ crosses the unity line.
- The theoretical implications and context need more interesting and accessible treatment if this is to be more than a methods paper, and if people are going to start using these tasks as their solution to measuring reliable individual differences. As the authors point out in discussion, increasing reliability does not necessarily mean you have increased the ability to measure the thing you set out to measure (validity). Clearly the reliability of the tasks comes partly from making them more difficult, which is fine as long as the difficulty is coming from the thing you wanted to measure – in this case, ability to overcome conflict, rather than, for example, memory load for the task rules (unless you have a theoretical framework in which memory is a critical component of conflict resolution). Does it matter that the tasks combine different types of conflict that may have different origins? (personally I think it is pragmatic to combine them as a way forward, given that the 'purer' tasks don't currently offer enough reliability to be able to productively use them for individual differences research).
- The statement on split-half vs test-retest reliability is too mild. In nearly every application I can think of, it is really test-retest reliability that matters, given that researchers will want trait scores that represent cognitive abilities that do not vary with the specific day of testing. As such, the need for test-retest reliability is implicit in the word trait, and so we could question if the use of the word trait in the current analysis is the right label without test-retest data.

Reviewer #3 (Remarks to the Author):

The status of executive function (EF) as a psychological construct, and inhibitory control in particular, is a mess because the measures show little convergent validity and interesting questions about its relationship to other constructs are inconsistent because coherent latent variables are difficult to extract. This set of tasks developed and tested in this study are a possible godsend, but I have some questions and requests. Researchers will want to use this set of tasks only if they generate a coherent latent variable. Do they? One of the best tasks in the set is a standard flanker that based on the literature and the current analysis should have poor reliability. It is not combined with Simon (location) conflict. It is not followed by a second task. There is no explanation for why this instantiation of the flanker should be better than that used in the Hedges et al. study or the many others that show poor convergent validity. The current version does jitter the horizontal location of the arrows, but having some degree of location uncertainty is not unusual. We really need to know why

the flanker task works as well, if not better, than interference tasks that are really quite novel. Examination of Figure 1 c and d provide a possible explanation assuming that these are screen shots of the materials actually used in the experiments. The "arrows" are not actually arrows, but look like this: >><>. The incongruent trials form an object <> that has a pop out quality that must then be unpacked in order to determine the direction of the center target. If this wild speculation is actually the answer to why this specific version of a standard flanker task works (is reliable), it needs to be verified. Finally, the authors should be commended for making their tasks and data available. Ken Paap

Reviewer #4 (Remarks to the Author):

This paper addresses an important and timely issue in experimental psychology, and the behavioral sciences more generally. The "reliability paradox" of stable group-level effects failing to translate to measurement of individual differences poses serious challenges for researchers seeking task-independent measurement of general cognitive capacities. As noted by the authors, recommendations for how to address poor reliability have been wide-ranging, with some advocating for abandoning response time measures altogether (e.g., Draheim et al., 2019) and others recommending more conventional (but potentially problematic) solutions (e.g., increased trial numbers; Rouder et al., 2019).

The authors explore alternative methodological and analytic approaches of addressing the reliability paradox. The authors consider the standard versions of conflict tasks, as well versions that increase complexity at the task level (e.g., via hybrid tasks that combine elements of the Stroop and Simon), at encoding/response levels (e.g., the "double-shot" versions of the task that require more extensive stimulus encoding as well as multiple responses on some trials), as well as in terms of visual complexity (i.e., via "gamification" of the stimulus display). They then report a hierarchical Bayesian analysis that allows them to quantify the ratio of "trait" variability (in the conflict effect) to variability due to noise in both conventional versions of the conflict tasks as well as the more complex versions of those tasks.

The key findings are that the more complex versions of the tasks are markedly more efficient than their traditional counterparts in terms of the number of trials required to attain conventionally acceptable levels of reliability (e.g., with 50% fewer trials, sometimes an even greater reduction). The analysis further reveals that the more complex task variants achieve this by effectively boosting the conflict signal in the data, eliciting larger conflict effects than are typically found in experimental settings with more stripped down versions of the tasks.

My overall impression of the manuscript is highly positive. As mentioned above, the work addresses a timely problem that has broad theoretical and practical implications for psychology and the behavioral sciences. I have only very minor comments, all of which could be addressed in a revision.

1 – While the authors very nicely discussed why the more complex and gamified versions of the conflict tasks would likely improve the reliability of the tasks by increasing the magnitude of the conflict effects, I found the impact to be lower than it could be. In some places, the authors provide more detailed points on why conflict effects could be enhanced in the non-standard versions of the tasks (e.g., people can't blur their vision to prevent effective processing of the text in the Stroop), but it is not always easy to appreciate the benefits over the standard laboratory tasks.

A compact tabular summary of the specific advantages offered by the more complex tasks over the standard ones would be extremely useful for the reader to appreciate why the additional complexities are desirable. Opting for more complex tasks goes against the standard practice of how experimental psychologists put great effort into distilling a task to its simplest (and often sparsest) presentation to isolate a mechanism of interest. Discussing how this approach can be counterproductive—as the

authors demonstrate with respect to conflict tasks—warrants more discussion. Are there other areas of experimental psychology that might benefit from increased task complexity? The authors note prospective memory as one alternative, but presumably there are others.

2 – The reanalysis of the Hedge et al. data could have been better signposted. Discussing the purpose of the analysis in a more up-front manner would better contextualize it and the analysis of the more complex tasks. A sentence or two stating the goals of the reanalysis would be enough here (e.g., establish baseline levels of η in the standard task, etc.).

3 – The model comparisons with the no-practice and conflict \times practice interaction models are important, but not explicitly motivated. A sentence or two clarifying why comparisons between the standard model and these alternatives might be important would clarify the purpose of this aspect of the analysis (e.g., allows examination of the block-wise effectiveness of the tasks in eliciting conflict effects; relevant to the discussion of fatigue with larger trial numbers).

4 – Typo Line 159 – “more likely that the standard model”

5 – Some issues with the axis label symbols not rendering correctly in the Figures. Could column headers be used to label the data being presented instead?

6 – Line 382 – How slow did RT have to be to trigger a “Hurry Up” prompt?

7 – Typo Line 386 – “errors where required”

Response to Reviewer comments

We thank all Reviewers for their comments, suggestions, and queries.

Reviewer #1

In the present manuscript, the authors propose a solution to the reliability paradox – the general finding that tasks which produce robust and reliable group effects tend to be unreliable in differential research). Their solution entails administering gamified versions of popular cognitive conflict tasks (Stroop, flanker, and Simon, which are known to be good examples of the reliability paradox), including combining Simon tasks with the other two tasks, designed to obtain larger RT interference effects and therefore increased reliability. Using hierarchical analysis, the authors found that these gamified and combined measures generally demonstrate higher reliability, some of which in under 100 trials.

This manuscript has a number of strengths, including that it concerns an important topic in individual differences and is well-written.

>> Thanks!

There have been many studies using various methods over the last several years trying to find solutions to the reliability paradox and most of them, frankly, fail to show significant improvements. I also highly appreciate that the authors made their tasks available to other researchers and encouraged their use.

>> Thanks

That said, this manuscript is quite methodological and psychometric in nature, and so I believe it would more appropriate for an experimental or methodological journal than Nature Communications.

>> Our revision follows your guidance to make the manuscript more appropriate for Nature Communications.

I also believe there are some limitations of the current research, and important discussion of these (which would likely increase the impact of this paper considerably) is rendered impossible given the word requirements of Nature Communications. Given these concerns, my recommendation would be to reject this manuscript. However, I think there are merits to this study and that it does advance the field of science. I think it would be a much better fit in a different journal – one that would allow more space to give the methods and results more discussion than is currently possible. In short, I think this manuscript is not well suited for publication in a multi-disciplinary with strong length requirements. Below is a list of my comments and concerns.

>> Fortunately, our initial submission was well under the word limit (by almost 2000 words in the main body, 1200 words in the methods, with space for 7 extra figures and tables and ~30 references). We have used this extra space to address the excellent feedback you and other reviewers have offered in order to improve the paper.

Major comments and concerns:

1. As the authors note on p. 10 lls 233-234, this work does not address the question as to whether validity is improved when using these more reliable measures. This is a major limitation. I am very skeptical that merely improving the reliability of these tasks would result in appreciable gains in terms of validity, especially when sticking with RT contrasts.

>> Our aim is not to claim gains in validity regarding the tasks we study because their validity in terms of the specific types of interference control that they measure is well established in the experimental literature over many decades as we now discuss. We believe that the reviewer may be using “validity” to refer to measurement of a more domain-general construct or construct through correlational approaches. We have revised the paper to avoid this confusion, making clear that we do not make any claims about “validity” of this type. The existence of such domain-general control constructs is controversial, as is the use of correlation-based approaches to establish them. We have added a discussion of this controversy, and note that although it is necessary to have reliable measures to pursue the correlational approach, it is also important to be cautious about such evidence given the well-established and numerous critiques.

Improving the reliability of conflict tasks is an important step, but there are quite a few ways this can be done in a way that is not theoretically interesting in terms of assessing the underlying mechanisms and cognitive processes believed to be involved here. For example, Hedge et al. (2021) demonstrated that conflict tasks have very little validity for assessing cognitive conflict. This is not only a reliability problem, but, as they noted, even if reliability was improved and strong correlations were found among conflict tasks and various other measures, they would likely be attributable to factors such as response cautiousness and/or processing speed (i.e., not attention, inhibition, or conflict resolution).

Without a proper validation of these new measures, I would assume something similar here. Studies over and over have shown that even with (somewhat) reliable versions of conflict tasks, there is still little-to-no correlation with other executive functioning tasks when using difference scores, suggesting that the problem with these measures is not only that difference scores are involved in calculation of performance (see Draheim et al., 2019; also see Rey-Mermet et al., 2019 who noted that difference scores appear to not be as process pure as advertised).

>> These papers refer to a general control construct. As previously stated, we make no claims about validity in that regard.

This is also related to the argument from Rouder and colleagues that, in short, these tasks simply do not appear to measure what they are thought to from the correlational perspective.

>> We are unsure what papers are being referred to and so we have discussed the matter with Jeff Rouder and Julia Haaf. They were clear that they did not question the validity of conflict tasks in any of their papers, only whether they correlate with each other indicating the presence of some general factor, and whether this was due to a lack of reliability or to there being no such general factor.

Without any sense of validity (including criterion validity and discriminant validity), it is not possible to know whether the observed improvements in reliability are theoretically meaningful.

>> We previously cited summaries (in the Introduction) of the very large experimental literature validating these tasks as measures of control of specific types of interference (linguistic, spatial, visual) and have now more strongly emphasised what they have shown.

2. Related to the above point; P. 4 lls 98-99 the authors state in regard to abandoning RT difference scores that "...this potentially discards advantages of difference scores in terms of validity." This is something deserving of more discussion because it is central to the hypotheses and approach the authors used.

>> As above, we have now done this (e.g., p. 3, Introduction).

The authors are operating with the hypothesis that improving the reliability and effect sizes in these tasks will improve their validity. The authors need to justify their reason for sticking with RT difference scores to measure performance in these tasks, despite a large body of evidence showing RT differences are, not just often unreliable, but display poor criterion-related validity.

>> We are not operating on such a hypothesis and have tried to be clear about that in the revision by explicitly addressing validity.

There is ongoing debate as to whether RT difference scores are as process pure as many believe, with researchers increasingly recognizing that the problems with difference scores (primarily in RT) and conflict tasks more generally are not isolated to unreliability but also validity (e.g., Draheim et al., 2019; Hedge et al., 2021; Rey-Mermet et al., 2019).

>> These papers address only validity with respect to the controversial idea of domain general control, they do not provide evidence against validity with respect to specific types of interference.

Note in the developmental/aging literature, it has long been recognized that RT Stroop effects are artificially increased in individuals with slower processing speed.

>> We agree that RT effect differences are typically proportional to overall RT. This issue has long been addressed in the aging literature by analysing log RT, which we have used in our analyses. Our Results section now explicitly points out this advantage of our analysis so that readers can clearly see that our analysis has resolved this reviewer's concern.

A point I have made before is that one of the issues with the logic of difference scores in the context of conflict tasks is that there is evidence that even congruent trials require controlled processing, to an extent, especially when presented in mixed-blocks designs as is often the case (in the tasks switching literature, this is known as a mixing cost.) This is particularly true when congruent trials are not more frequent than incongruent trials.

As one example, see Figure 2 from Heitz & Engle (2007) using the Eriksen flanker task (attached to this review). Notice how high vs. low working memory capacity individuals differ in their RT on compatible (congruent) trials. Using difference scores removes this source of individual variation.

Other studies have also found evidence of controlled processing in the baseline trials of various attention measures – including prosaccade trials of antisaccade tasks (Unsworth et al., 2004) and congruent trials in color Stroop (Kane & Engle, 2003).

>> Thanks for pointing these out, this is an important point that we now address in the Introduction and expand on with other evidence that does not rely on the assumption about working memory capacity.

3. The research presented in this manuscript is highly methodological in nature, using advanced statistics many researchers are not familiar with. There are also multiple experiments, multiple unique tasks and task combinations, comparisons to previous research, etc. The complexities of the study are difficult, if not impossible, to properly convey to the reader in a journal with such limited page space. I submit that it would be beneficial to both the authors and the reader to have a detailed introduction and explanation of methodology and results (including many more tables and/or figures), which is not possible in Nature Communications.

>> We have included a detailed introduction and explanation of the methodology and results (including more display items), taking advantage of the extra space available.

To this point, I found it difficult to evaluate the results because it was boiled down to just a couple pages and figures, and without mention of individual reliability estimates,

>> These were provided, but perhaps not in the format that the reviewer is familiar with. We have attempted to clarify our findings in terms of both written and graphical explanations.

correlations between tasks (if some of the same tasks were administered to the same participants – was not entirely clear), and so on.

>> As we originally said (and have now clarified further) the tasks were administered to different participants.

At points, I had to mostly take the authors' word regarding what they found, because I did not have access to a table or individual figures to interpret the results for myself.

>> We provided complete data and open-source code with all information that would be required to make a detailed assessment. Again, we have sought to increase the clarity of our writing and figures. We would be happy to make further amendments in line with specific requests from the reviewer.

A longer manuscript could more thoroughly describe not just the results and the different experiments (with additional tables and figures for important aspects of the data omitted from the present manuscript, presumably due to length limits), but expand on practical recommendations, limitations, and other important factors which would allow the reader to better digest this research, and perhaps improve the impact of this line of work.

>> As well as more detailed Results, we have expanded on the practical recommendations and limitations in the Discussion.

4. The comparison to results from Hedge et al. (2018) strikes me as not entirely appropriate, i.e., somewhat limited. It is useful to have a comparison to non-gamified versions of these tasks, but comparisons across labs and samples have limitations. For example, it is not possible to know whether non-gamified versions of these tasks given to the participants from the present experiments would also have showed larger effects and better reliability.

>> An experiment testing non-gamified versions of our most promising tasks has been conducted, with the results added to the manuscript. We also cover this more thoroughly in the Discussion.

5. These data seem to be based on a very small group of subjects. The overall sample size of 181 participants for the final experiment is just barely enough for a validation study (i.e., enough to obtain stable correlation estimates; see Schönbrodt & Perugini, 2013), but the authors note on p. 14 that there were only 30 participants per condition. It is not entirely clear to me what a condition is in the context of this study (30 per task?). Either way, the authors need to make additional steps to justify

how such a small sample size is appropriate for a validation study attempting to solve the reliability paradox.

>> The cited recommendation is for correlation-based methods, which we did not employ because, as we explained earlier, our aim was not to test whether there is a single inhibition construct or other type of general factor. This comment also ignores the very large samples in earlier experiments with low numbers of trials per participant, which produced consistent results with the final experiment, which importantly has high data quality in the sense that participants completed many trials. As we referenced in the original manuscript, Smith and Little (2018) demonstrate the importance of data quality. Further, the extra non-gamified experiment essentially serves as a replication that effectively doubles the sample size for the high-quality data. Hence, we believe these concerns are unfounded. We have emphasised these points in the paper to avoid confusion.

Minor comments and concerns:

1. On p.3 ll 63, the authors state that the reliability paradox was recently identified by Hedge et al. (2018). This is not quite accurate.

The term “reliability paradox” was coined by Hedge et al. (2018) in a fantastic and now seminal paper, but the paradox had been identified long before Hedge et al., dating at least as far back as the 50s-70s with psychometricians such as Lee Cronbach and Fred Lord, among others, debating how to measure change. The paradox was also discussed by Overall and Woodward (1975) in a paper entitled “Unreliability of Difference Scores: A Paradox Measurement of Change”, in which they noted that unreliability of difference scores is a concern for correlational research but argued that it was not a problem for experimental research because statistical power is actually increased at low reliability (and maximized at a reliability of 0), and by Logie et al. (1996) who noted that robust experimental effects do not translate to reliability at the individual level.

In regard to conflict tasks such as Flanker, Simon, and Stroop (and task-switching tasks), the reliability paradox was discussed (but not named) by Draheim et al. (2016), although the focus of that paper was more so the lower reliability of task-switching measures (and, again, this was based on previous research, I am by no means suggesting we identified the paradox). Friedman and Miyake (2004) also noted that difference scores were a likely cause of the low reliability and validity they found in measures of inhibition, including Stroop and flanker. There are a number of other papers that touch on the paradox by researchers (i.e., in the clinical or developmental space) who understand and appreciate the fundamental difference between experimental and differential research.

To summarize, while Hedge et al. named the “reliability paradox”, it is not an entirely accurate characterization that they were the first to identify it, either generally or in regard to conflict tasks. The authors should consider modifying their introduction accordingly.

>> Thanks very much for this scholarly summary, we have used it to improve the attribution in the paper.

2.P. 5 116-119 the authors note that simply adding more trials to the task is likely not a great approach, and one reason being that fatigue or practice effects can influence reliability. This is a really great point. It might be worth mentioning a collaboration project (preprint) that counterintuitively found no correlation between number of trials and reliability of tasks using difference scores (analysis was based on a systematic review involving 46 studies, see p. 33 of von Bastian et al., 2020).

>> Thank you, we have now cited this paper in the Introduction.

Also important is that while Rouder et al. noted increasing trials can improve reliability, they also were skeptical of this approach being successful given it would take possibly as many as 1000 trials (or possibly more) to reach adequate reliability in some conflict tasks.

>> We agree, this was exactly the point we already made in the paper, thus, we have tried to be clearer about this.

-Chris Draheim

Reviewer #2 (Remarks to the Author):

The paper addresses a topic that is certainly of wide interest – offering a solution to the difficulties of measuring individual differences in cognition (e.g. as evidence of cross-disciplinary interest, the paper whose data is used in figure 2, Hedge et al. 2018, has 700 citations, and that paper only highlighted and explained the problem rather than offering a solution).

The current paper makes the following valuable contributions:

- Shows that combining conflict tasks can provide more reliable individual differences in a feasible number of trials for many applications.
- Uses a more engaging gamified environment for the tasks, which likely contributes to the reliability by keeping participants on task for longer.
- Provides the code for these tasks openly and freely
- Deploys sophisticated analysis which, if I understand correctly, both helps the reliability of cognitive individual differences by handling practice effects and generally smoothing through the model fitting process, and also helps understand the nature of the data through separating ‘trait’ variance and measurement variance.

Thus overall, the paper puts the measurement of cognitive individual differences on a more positive footing, which is very much needed. It gives researchers something to work with, despite the limitations and suggestions listed below.

>> Thanks! Although we agree it is likely that gamification increases engagement, we note that the new non-gamified experiment indicates this was not necessary to improve reliability, at least in our sample of MTurk participants.

However, there are several ways I would suggest the paper can be improved if it is to be published in a general audience journal.

- The accessibility of the text will prove challenging for most readers in many sections.

>> We have attempted to improve accessibility for a general audience using the extra space available to us. This includes providing greater detail on our tasks and outcomes, as well as expanding our discussion of many points throughout.

- The accessibility of the figures also needs to be improved, even for specialist readers. Try to avoid acronyms as axis labels, because it makes a reader's job so much harder to translate them all. Most importantly, the most critical message of the paper is in column 6 of the figures, but it is extremely difficult to understand what these plots are plotting and the rationale for it. The y axis of $n=2xL$ is not explained in the main text or in the figure legend, and nor is why the critical number of trials is where trials vs $n=2xL$ crosses the unity line.

>> We have broken up the large figure and made the changes suggested in terms of labels.

- The theoretical implications and context need more interesting and accessible treatment if this is to be more than a methods paper, and if people are going to start using these tasks as their solution to measuring reliable individual differences. As the authors point out in discussion, increasing reliability does not necessarily mean you have increased the ability to measure the thing you set out to measure (validity). Clearly the reliability of the tasks comes partly from making them more difficult, which is fine as long as the difficulty is coming from the thing you wanted to measure – in this case, ability to overcome conflict, rather than, for example, memory load for the task rules (unless you have a theoretical framework in which memory is a critical component of conflict resolution).

>> As suggested by other reviewers, we have now addressed the issue of validity more thoroughly, especially in the Discussion. We note the need for future work regarding the issue of whether these tasks provide valid measurements of a single factor related to the ability to overcome a broad array of different types of conflict. We also note that the improved reliability of our tasks is a foundational step towards the latter aim.

Does it matter that the tasks combine different types of conflict that may have different origins? (personally I think it is pragmatic to combine them as a way forward, given that the 'purer' tasks don't currently offer enough reliability to be able to productively use them for individual differences research).

>> We addressed this briefly in the original paper and have now expanded the discussion, and also emphasised the point about pragmatics.

- The statement on split-half vs test-retest reliability is too mild. In nearly every application I can think of, it is really test-retest reliability that matters, given that researchers will want trait scores that represent cognitive abilities that do not vary with the specific day of testing. As such, the need for test-retest reliability is implicit in the word trait, and so we could question if the use of the word trait in the current analysis is the right label without test-retest data.

>> We have strengthened the statement in line with the reviewers statements, but believe it is justifiable to use "trait" in the current analysis with greater emphasis on the point about time scale, given the continuum involved here (i.e., stability in a session, over days, weeks, months etc.). We note this in the Discussion.

Reviewer #3 (Remarks to the Author):

The status of executive function (EF) as a psychological construct, and inhibitory control in particular, is a mess because the measures show little convergent validity and interesting questions about its relationship to other constructs are inconsistent because coherent latent variables are difficult to extract. This set of tasks developed and tested in this study are a possible godsend, but I have some questions and requests.

>> Thanks!

Researchers will want to use this set of tasks only if they generate a coherent latent variable. Do they?

>> In line with comments from other reviewers we have clarified that our aim was not to answer this question, and hence our design was not appropriate to do so, although making reliable tasks available is a pre-requisite for exploring this question. However, although we acknowledge that forming a coherent latent variable would be convenient, if this turns out not to be the case, we think these tasks remain important for understanding the multi-faceted nature of conflict control, and we have now further elaborated this point.

One of the best tasks in the set is a standard flanker that based on the literature and the current analysis should have poor reliability. It is not combined with Simon (location) conflict. It is not followed by a second task. There is no explanation for why this instantiation of the flanker should be better than that used in the Hedges et al. study or the many others that show poor convergent validity. The current version does jitter the horizontal location of the arrows, but having some degree of location uncertainty is not unusual. We really need to know why the flanker task works as

well, if not better, than interference tasks that are really quite novel. Examination of Figure 1 c and d provide a possible explanation assuming that these are screen shots of the materials actually used in the experiments. The "arrows" are not actually arrows, but look like this: >><>. The incongruent trials form an object <> that has a pop out quality that must then be unpacked in order to determine the direction of the center target. If this wild speculation is actually the answer to why this specific version of a standard flanker task works (is reliable), it needs to be verified. Finally, the authors should be commended for making their tasks and data available. Ken Paap

>> This "wild speculation" is actually a great point, which we now discuss! Although we agree that jitter is not unusual, we did determine during the earlier experiments that having sufficient magnitude (two character widths rather than just one) helped. We believe the issue is now further clarified, although not solved, by our finding of the same good performance in non-gamified versions, strengthening the argument that the display itself is important.

Reviewer #4 (Remarks to the Author):

This paper addresses an important and timely issue in experimental psychology, and the behavioral sciences more generally. The "reliability paradox" of stable group-level effects failing to translate to measurement of individual differences poses serious challenges for researchers seeking task-independent measurement of general cognitive capacities. As noted by the authors, recommendations for how to address poor reliability have been wide-ranging, with some advocating for abandoning response time measures altogether (e.g., Draheim et al., 2019) and others recommending more conventional (but potentially problematic) solutions (e.g., increased trial numbers; Rouder et al., 2019).

The authors explore alternative methodological and analytic approaches of addressing the reliability paradox. The authors consider the standard versions of conflict tasks, as well versions that increase complexity at the task level (e.g., via hybrid tasks that combine elements of the Stroop and Simon), at encoding/response levels (e.g., the "double-shot" versions of the task that require more extensive stimulus encoding as well as multiple responses on some trials), as well as in terms of visual complexity (i.e., via "gamification" of the stimulus display). They then report a hierarchical Bayesian analysis that allows them to quantify the ratio of "trait" variability (in the conflict effect) to variability due to noise in both conventional versions of the conflict tasks as well as the more complex versions of those tasks.

The key findings are that the more complex versions of the tasks are markedly more efficient than their traditional counterparts in terms of the number of trials required to attain conventionally acceptable levels of reliability (e.g., with 50% fewer trials, sometimes an even greater reduction). The analysis further reveals that the more complex task variants achieve this by effectively boosting the conflict signal in the data, eliciting larger conflict effects than are typically found in experimental settings with more stripped down versions of the tasks.

My overall impression of the manuscript is highly positive. As mentioned above, the work addresses a timely problem that has broad theoretical and practical implications for psychology and the behavioral sciences.

>> Thanks!

I have only very minor comments, all of which could be addressed in a revision.

1 – While the authors very nicely discussed why the more complex and gamified versions of the conflict tasks would likely improve the reliability of the tasks by increasing the magnitude of the conflict effects, I found the impact to be lower than it could be. In some places, the authors provide more detailed points on why conflict effects could be enhanced in the non-standard versions of the tasks (e.g., people can't blur their vision to prevent effective processing of the text in the Stroop), but it is not always easy to appreciate the benefits over the standard laboratory tasks.

>> As requested by other reviewers we have expanded our treatment of this point (in the Introduction and Discussion).

A compact tabular summary of the specific advantages offered by the more complex tasks over the standard ones would be extremely useful for the reader to appreciate why the additional complexities are desirable.

>> We have added further clarification in the paper rather than the suggested table as some of the nuance was lost in this format. Hopefully the additional written detail will suffice. However, this prompted us to include Table 1 as a way to help readers understand what was included in each of the experiments.

Opting for more complex tasks goes against the standard practice of how experimental psychologists put great effort into distilling a task to its simplest (and often sparsest) presentation to isolate a mechanism of interest. Discussing how this approach can be counterproductive—as the authors demonstrate with respect to conflict tasks—warrants more discussion. Are there other areas of experimental psychology that might benefit from increased task complexity? The authors note prospective memory as one alternative, but presumably there are others.

>> Good point. Where tasks are simplified for control, they also reduce cognitive load and this may be why some of our most effective tasks go against the general rule of parsimony. We have provided further detail of the prospective memory example but are not aware of other examples. We also discuss this issue with respect to the new non-gamified experiments, which seems to indicate the associated task complexity was not necessary for increased reliability.

2 – The reanalysis of the Hedge et al. data could have been better signposted. Discussing the purpose of the analysis in a more up-front manner would better contextualize it and the analysis of the more complex tasks. A sentence or two

stating the goals of the reanalysis would be enough here (e.g., establish baseline levels of eta in the standard task, etc.).

>> We have added the requested signposting (at the start of the Results).

3 – The model comparisons with the no-practice and conflict × practice interaction models are important, but not explicitly motivated. A sentence or two clarifying why comparisons between the standard model and these alternatives might be important would clarify the purpose of this aspect of the analysis (e.g., allows examination of the block-wise effectiveness of the tasks in eliciting conflict effects; relevant to the discussion of fatigue with larger trial numbers).

>> These are great points and we have added the requested clarifications.

4 – Typo Line 159 – “more likely that the standard model”

5 – Some issues with the axis label symbols not rendering correctly in the Figures. Could column headers be used to label the data being presented instead?

6 – Line 382 – How slow did RT have to be to trigger a “Hurry Up” prompt?

7 – Typo Line 386 – “errors where required”

>> These have been fixed. For point 6, this applied to RTs > 2 seconds.

REVIEWERS' COMMENTS

Reviewer #1 (Remarks to the Author):

I appreciate the authors' efforts to address concerns of all reviewers and think they have done a fantastic job of it in the revised manuscript and author response. My initial recommendation was to reject due mostly to journal fit concerns, however after assessing the revision, I am comfortable recommending acceptance in Nature Communications because:

1) The authors had leftover space in the original submission for text and figures and so they were able to address reviewer concerns in this space.

2) The authors were quite convincing in their arguments against my initial, and most pressing, concern that the study did not include validation of these tasks as measures of a domain-general attention construct.

As for the Rouder et al. comment, I had distinctly remembered a line from the original PsyArXiv pre-print of Rouder and Haaf (2019) along the lines of, "Simply put, these tasks do not appear to measure what they are widely believed to." in regard to the Stroop and flanker tasks. I went back to the published paper to look for it after reading the author response letter and could not find the quote, so it's entirely possible I am misremembering, confusing this paper up with another, or maybe there was a similar quote in the pre-print that did not make it to the published version.

-Chris Draheim

Reviewer #2 (Remarks to the Author):

The Authors have addressed all my suggestions in the revision. I have no further comments.

I believe this paper will make a very positive contribution to the field.

Reviewer #3 (Remarks to the Author):

The authors were very responsive to my questions and comments, primarily in their cover letter. I understand that their goal was to improve reliability and leave the question of enhanced validity (especially for a domain-general inhibitory-control mechanism for future work. I also acknowledge that they added some material to the introduction supporting the idea that these traditional nonverbal interference tasks have theoretical implications in their own right that will be enhanced by having more reliable versions of these traditional versions. I concede that this may be true, but I still suspect that wide-spread use of these new tasks will not occur until it is shown that they enjoy convergent validity.

I am still surprised that the "plain" flanker task is the best (most reliable) task in this new set. This seems to undermine most of the rationale for why the overall set is more reliable than the Hedge's set and most of the published work using the flanker task. If the active ingredient in the present flanker is a <> pop-out effect, it would be worthwhile to include a test of this hypothesis.

Although the authors have tried very hard to describe their work within the structure of a Nature paper, it is extremely difficult to read and understand. I think it will be penetrated only by the most highly motivated readers with high domain expertise.

Reviewer #4 (Remarks to the Author):

The authors have addressed all of the concerns I raised in my initial review. I appreciate the expanded discussion regarding standard vs. non-standard versions of the conflict tasks. Further, the comparison of gamified vs. non-gamified versions of the tasks highlights the likely importance of motivation to complete the task (i.e., engagement) as an important factor.

Response to Reviewer comments

We, again, thank the Reviewers for all of their comments and feedback on this manuscript.

Reviewer #1 (Remarks to the Author):

I appreciate the authors' efforts to address concerns of all reviewers and think they have done a fantastic job of it in the revised manuscript and author response. My initial recommendation was to reject due mostly to journal fit concerns, however after assessing the revision, I am comfortable recommending acceptance in Nature Communications because:

1) The authors had leftover space in the original submission for text and figures and so they were able to address reviewer concerns in this space.

2) The authors were quite convincing in their arguments against my initial, and most pressing, concern that the study did not include validation of these tasks as measures of a domain-general attention construct.

>> Thank you for the positive review.

As for the Rouder et al. comment, I had distinctly remembered a line from the original PsyArXiv pre-print of Rouder and Haaf (2019) along the lines of, "Simply put, these tasks do not appear to measure what they are widely believed to." in regard to the Stroop and flanker tasks. I went back to the published paper to look for it after reading the author response letter and could not find the quote, so it's entirely possible I am misremembering, confusing this paper up with another, or maybe there was a similar quote in the pre-print that did not make it to the published version.

-Chris Draheim

Reviewer #2 (Remarks to the Author):

The Authors have addressed all my suggestions in the revision. I have no further comments.

I believe this paper will make a very positive contribution to the field.

>> Thanks for the kind feedback.

Reviewer #3 (Remarks to the Author):

The authors were very responsive to my questions and comments, primarily in their cover letter. I understand that their goal was to improve reliability and leave the question of enhanced validity (especially for a domain-general inhibitory-control mechanism for future work. I also acknowledge that they added some material to the

introduction supporting the idea that these traditional nonverbal interference tasks have theoretical implications in their own right that will be enhanced by having more reliable versions of these traditional versions. I concede that this may be true, but I still suspect that wide-spread use of these new tasks will not occur until it is shown that they enjoy convergent validity.

>> Thanks, we agree that correlation-based approaches to validity can be important with respect to some applications of measurements from these tasks, and have noted this in the manuscript.

I am still surprised that the "plain" flanker task is the best (most reliable) task in this new set. This seems to undermine most of the rationale for why the overall set is more reliable than the Hedge's set and most of the published work using the flanker task. If the active ingredient in the present flanker is a <> pop-out effect, it would be worthwhile to include a test of this hypothesis.

>> We agree that there needs to be more systematic study of the way variations in Flanker displays modulate reliability. We now note both that a great variety of different displays have been used (so we are not sure that there is necessarily a "plain" Flanker task), that Hedge et al.'s low reliability results were associated with a display that has little apparent grouping (as arrows were arrayed vertically), and have called for more systematic study but have not done so ourselves given we now have little extra room to play with.

Although the authors have tried very hard to describe their work within the structure of a Nature paper, it is extremely difficult to read and understand. I think it will be penetrated only by the most highly motivated readers with high domain expertise.

>> Thanks for the feedback.

Reviewer #4 (Remarks to the Author):

The authors have addressed all of the concerns I raised in my initial review. I appreciate the expanded discussion regarding standard vs. non-standard versions of the conflict tasks. Further, the comparison of gamified vs. non-gamified versions of the tasks highlights the likely importance of motivation to complete the task (i.e., engagement) as an important factor.

>> Thank you for the positive comments.